



# Temporal evolution of crack propagation characteristics in a weak snowpack layer: conditions of crack arrest and sustained propagation

**Bastian Bergfeld[1], Alec van Herwijnen[1], Grégoire Bobillier[1], Philipp L. Rosendahl[2], Philipp Weißgraeber[3], Valentin Adam[2,1], Jürg Dual[4] and Jürg Schweizer[1]**

[1] WSL Institute for Snow and Avalanche Research SLF, Davos, Switzerland

[2] Institute of Structural Mechanics and Design, Technical University of Darmstadt, Darmstadt, Germany

[3] Chair of Lightweight Design, University of Rostock, Germany

[4] Institute for Mechanical Systems, ETH Zürich, Zürich, Switzerland

*Correspondence to*: Bastian Bergfeld (bastian.bergfeld@slf.ch)

**Abstract.**

*For a slab avalanche to release, the system, consisting of a weak layer below a cohesive slab, must facilitate crack propagation over large distances – a process we call dynamic crack propagation. Field measurements on crack propagation at this scale are very scarce. We therefore performed a series of propagation saw test experiments, up to ten meters long, over a period of*

*10 weeks and analyzed these using digital image correlation techniques. We derived the elastic modulus of the slab (0.5 to 50 MPa), the elastic modulus of the weak layer (50 kPa to 1 MPa) and the specific fracture energy of the weak layer (0.1 to 1.5 J m$^{-2}$) with a homogeneous and a layered slab model. During crack propagation, we measured crack speed, touchdown distance and the energy dissipation due to compaction and dynamic fracture (5 mJ m$^{-2}$ to 0.43 J m$^{-2}$). Crack speeds were highest for PSTs resulting in full propagation and crack arrest lengths were always shorter than touchdown lengths. Based on*

*these findings, an index for self-sustained crack propagation is proposed. Our data set provides unique insight and valuable data to validate models.*

## 1 Introduction

To better estimate avalanche release probability, we need to understand how the snow cover fails when a slab avalanche forms. Slab avalanche release is currently understood as resulting from a sequence of fracture processes. First, failure initiation and

crack growth to a critical size occurs before rapid crack propagation starts (onset of crack propagation). In the subsequent dynamic crack propagation phase, the crack self-propagates across the slope without the need of additional load. Avalanche release then occurs if the gravitational pull on the slab is great enough to overcome frictional resistance to sliding and to cause cracks at the crown, flank and stauchwall of the forming avalanche (Schweizer et al., 2003).

Avalanche size is a key component to predict the avalanche danger level (Meister, 1995;Statham et al., 2018;Techel et al.,

2020). There are essentially two processes that can limit avalanche release size: (1) either dynamic crack propagation stops (crack arrest) and the snow slab releases over the area where the weak layer is cracked, or (2) dynamic crack propagation





continues into flatter slope regions where subsequently the slab-bed friction prevents slipping of the detached slab (van Herwijnen and Heierli, 2009). In this case, the spatial extent of the avalanche is smaller than that of crack propagation. Spatial confinement of slopes and friction as limiting factors for avalanche size are quite intuitive whereas reasons for crack arrest in
unconfined slopes can be manifold. Unconfined slopes do not abut rock walls or dense forest and slab thickness does not taper to less than half the mean thickness (Jamieson and Johnston, 1992). Hence, changes in the snowpack are gradual, suggesting an intricate relation between fracture arrest and changes in snowpack properties. Simenhois and Birkeland (2014) investigated crowns and flanks of slab avalanches and suggested five controlling mechanisms for fracture arrest related to snowpack properties. Regarding the weak layer, they proposed fracture resistance and collapse height during fracture to be decisive. For
the slab, load (slab thickness and density), elastic modulus and tensile strength were considered as crucial. The latter is closely linked to slab tensile failure (slab fracture), an often-stated mechanism causing fracture arrest in the weak layer. Gaume et al. (2015) extended the mechanically based probabilistic model of Gaume et al. (2012) and Gaume et al. (2014) to analyse which snowpack properties influence slab fracture propensity. They concluded that stiff and thick snow slabs (high tensile strength) are less prone to slab fracture and therefore release larger avalanches, in line with field observations (van Herwijnen and
Jamieson, 2007). In addition to the tensile strength of the slab, an experimental and numerical study suggested that elastic modulus and density are relevant for crack arrest (Schweizer et al., 2014). More recently, Bobillier et al. (2022) performed discrete element simulations and reported elastic moduli of slab and weak layer and weak-layer shear strength influencing crack propagation propensity and crack speed. Furthermore, Simenhois and Birkeland (2014) encountered avalanches where they did not observe changes in snowpack properties at their boundaries and concluded that there may be other mechanisms
they did not identify.

In general, rapid crack propagation occurs when the instantaneous dynamic energy release rate $G$ equals the dynamic fracture energy $w_f^{dyn}$ of the weak layer (Freund, 1990). However, both quantities can be altered by the speed of the propagating crack. Thus, the speed of propagating cracks affects $w_f^{dyn}$ (Lee and Prakash, 1998) and is therefore a decisive parameter for crack arrest problems (Freund, 1990). For avalanches, this suggests that if the dynamically released energy of the slab falls below
the dynamic weak-layer specific fracture energy, crack propagation will halt and the size of the slab avalanche is determined. Despite the importance of crack speed in weak snowpack layers, there are not many substantiated crack speed values to be found in the literature. The first crack speed measurement in snow was reported by Johnson et al. (2004). They triggered a crack in flat terrain, measured snow surface velocity with seismic sensors, and determined a mean propagation speed of $20 \pm 2$ m s$^{-1}$ over a propagation distance of ~8 m. Since then, many studies estimated crack speeds from propagation saw tests
(PSTs), a fracture mechanical field experiment for snow (Gauthier and Jamieson, 2006;Sigrist and Schweizer, 2007). These studies used particle tracking velocimetry to measure slab displacement, from which crack speed was derived. In PSTs, crack speeds ranged from 10 to 50 m s$^{-1}$ (van Herwijnen and Jamieson, 2005;Birkeland et al., 2014;van Herwijnen et al., 2016). However, the typically 2–3 m long PST experiments are rather small tests, possibly not representative for slope scale crack propagation in avalanches. While the one-dimensionality of crack propagation along PST columns seems not be altering crack





propagation speed (Bergfeld et al., 2022), crack propagation in PSTs of typical size is affected by edge effects (Bergfeld et al., 2021;Bair et al., 2014).

At the slope scale, van Herwijnen and Schweizer (2011) derived a speed of $42 \pm 4$ m s$^{-1}$ over a distance of 60 m using seismic sensors deployed in an avalanche starting zone. To estimate crack speed, they determined the time difference between the first arrival of the signal generated by weak-layer fracture and the signal from the actual release of the avalanche, and assumed the

propagation distance to be the width of the avalanche. Based on videos of avalanches, Hamre et al. (2014) reported widely varying crack speed estimates ranging from 18 to 428 m s$^{-1}$. Bergfeld et al. (2022) improved the methodology and estimated crack speeds by evaluating 14 crack paths in an avalanche video recording, resulting in crack speeds between 23 and 44 m s$^{-1}$ (mean: $36 \pm 6$ m s$^{-1}$) covering distances from 26 to 440 m. In addition to these experimental studies, Trottet et al. (2022) performed numerical simulations based on the material point method (Gaume et al., 2018b) and reported the existence of a

transition from sub-Rayleigh anticrack to supershear crack propagation. While sub-Rayleigh anticrack propagation can explain crack speeds below a 100 m s$^{-1}$ the transition to supershear crack propagation (a crack propagating faster than the shear wave speed in the slab) potentially also explains the high crack speeds reported by Hamre et al. (2014).

While these studies are valuable initial contributions, a better understanding of the condition when the snowpack supports self-sustained crack propagation in the weak layer requires measuring crack speed and crack propagation distance in combination

with detailed snowpack properties. However, a data set with crack propagation speeds on scales larger than the typical 2 m long PSTs together with in-situ measured mechanical properties of the snowpack is not yet available. Furthermore, the assumption that crack speed is related to the propensity of a snowpack to promote self-sustained crack propagation stems from theoretical considerations, but experimental evidence is lacking.

The objective of this work is, first, to relate the crack speed in weak snowpack layers to the propensity of the snowpack to

promote self-sustaining crack propagation and, second, to provide an index for evaluating the propensity for self-sustaining dynamic crack propagation that can be derived from manual and simulated snowpack profiles.

Therefore, we conducted a series of very long PST experiments (up to 10 m long) over the entire life cycle of a weak layer, involving very different states of the snowpack. Highspeed videos of these experiments were analysed with image correlation techniques to estimate crack propagation characteristics, such as crack speed and crack propagation distance. In addition,

mechanical properties of the weak layer and slab, as well as specific fracture energy of the weak layer were estimated using two mechanical models. Furthermore, we suggest an estimate of dynamic fracture energy, which we call the dissipation of dynamic fracture, by separating the work done in the weak layer ahead of and behind the crack tip. In the end, our data set provides insight into which snowpacks promote self-sustained crack propagation and how propagation propensity can be estimated from measured or simulated snowpack profiles.


## 2 Methods

### 2.1 Field site and snowpack measurements

From 4 January to 19 March 2019, we performed 24 propagation saw tests (PST) on 22 field days on a flat and uniform site near Davos, Switzerland (Figure 1a), located on the roof of a building in a forest opening. In all PSTs, we tested the same weak layer, consisting of surface hoar ($\vee$, 10-15 mm) that had formed at the end of December 2018 (Figure 1b, red area). This layer of surface hoar was buried by consecutive snowfalls at the beginning of January 2019. Slab thickness increased over the measurement period (Figure 1b, blue line) and slab layers mainly consisted of fresh snow ($+$), decomposing and fragmented precipitation particles ($/$) and rounded grains ($\bullet$). On every field day, we characterized the snowpack with a manual snow profile following Fierz et al. (2009). To measure snow density, we used a 100 $cm^3$ cylindrical density cutter (38 mm diameter). To assess snowpack variations along the PSTs, we performed snow micro-penetrometer (SMP) measurements approximately every 50 cm. Generally, variations in penetration resistance along the PST column were small.

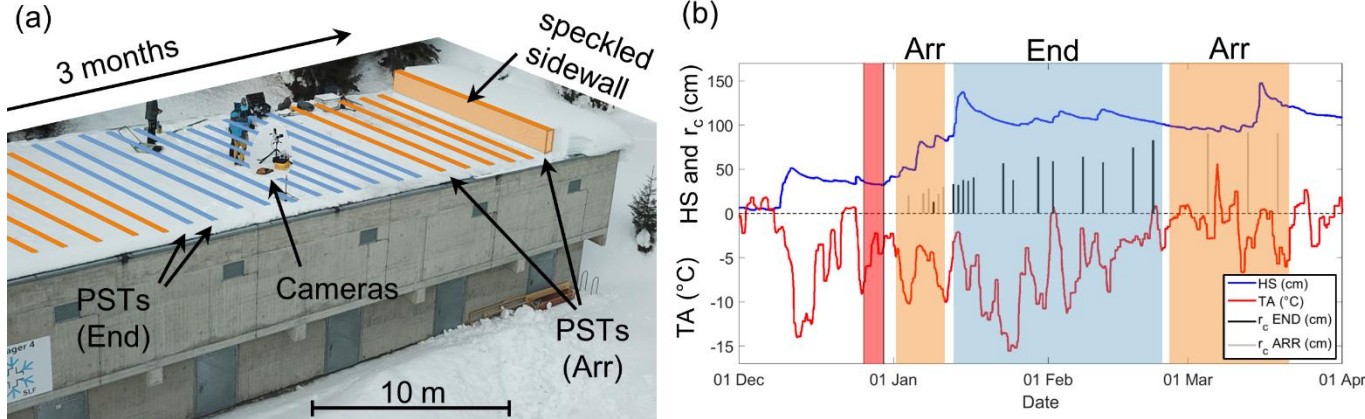

**Figure 1: (a) Field site where we performed numerous propagation saw tests (PSTs) resulting either in full propagation ("End", blue) or crack arrest ("Arr", orange). (b) Temporal evolution of snow height (HS), air temperature (TA) and critical cut length $r_c$. The tested weak layer consisted of surface hoar which had formed at the end of December (red region). After burial, PSTs initially resulted in crack arrest (orange background) before cracks propagated to the very end (blue background). In late February, air temperature (red line) increased above zero and PSTs resulted in cracks arrest again (orange). During the three months period, snow height (blue curve) increased and critical cut lengths (black and grey vertical lines) increased as well.**

### 2.1 PST experiments

One sidewall of each PST experiment was recorded with two cameras. We used a high-speed camera (Phantom, VEO710, 1280 pixel horizontal resolution) to investigate crack propagation across the PST, and we used a second camera (Sony RX100-V, 1920 x 1080 pixel², 50 fps) to capture slab deformation prior to crack propagation, when we cut the weak layer with a 2 mm thick snow saw. The side wall of the PST was speckled with black ink (Indian Ink, Lefranc & Bourgeois) to enhance the contrast for digital image correlation (DIC) analysis. Both cameras were aligned perpendicular to the centre-point of the PST beam. Camera distortion correction, DIC analysis and pre-processing (pixel to meter conversion, identifying slab, weak layer





and substratum) were done as described in Bergfeld et al. (2021). Camera settings (Table 4.A1) and settings for the DIC
analysis (Table 4.A2) are given in Appendix A.

The DIC analysis of the recorded high-speed videos provided us with the vertical ($w_{exp}$) and horizontal ($u_{exp}$) displacement
fields of the PST sidewall with time. Time resolution is given by the frame rate of the recordings (3000 – 22 000 for the high-
speed camera, see Table 4.A1). Spatial measurement resolution (6-27 mm) is given by the *step-size* of the subsets used for
DIC analysis (see Table 4.A2,) and the *pixel-to-meter* conversion factor (see Table 4.A1). In a further pre-processing step, we
calculated time derivatives from high-speed displacement fields to obtain velocity ($\dot{w}$, $\dot{u}$) and acceleration ($\ddot{w}$, $\ddot{u}$) of the slab.

**2.2 Location of the snow saw**

Prior to crack propagation, the location of the snow saw indicates the crack length which is required to determine the energy
release rate. Hence, crack length is a crucial parameter required for all frames. To accurately determine the position of the
snow saw, we therefore mounted a black dot onto the tip of the saw. In a post-processing step, we then went through all frames
to manually pick the location of the dot, since automated picking proved to be unreliable.

In a PST, it is difficult to keep the snow saw perfectly perpendicular to the direction of the sawing, and to saw at a continuous
pace. The saw often rotates horizontally, causing an offset between the dot on the saw tip and the actual location of the crack
tip. When rotating the saw back to perpendicular the dot on the saw tip rapidly accelerates while the crack tip almost remains.
We corrected the estimated saw locations for this error by first keeping the cut length constant at times the dot moved backward,
and we also smoothed the crack length $r(t)$ to avoid unnatural sharp kinks introduced from rotating the saw back to
perpendicular.

**2.2 Elastic modulus and specific fracture energy**

When cutting the weak layer in a PST, the unsupported slab behind the snow saw bends downwards due to its own weight.
Beam models can describe this bending behaviour when elastic parameters of slab and weak layer are known. Inversely, as
done by Bergfeld et al. (2021), the elastic parameters can be estimated by comparing measured horizontal ($u_{exp}$) and vertical
($w_{exp}$) displacement fields with predictions of a mechanical model (umod, wmod; Timoshenko beam resting on a Winkler
foundation; Rosendahl and Weissgraeber, 2020a). This is done by minimizing the residual ε between both fields:

$$\varepsilon = \sum_{k=0}^{N} \left| u_{mod}^{k}(E_{sl}, E_{wl}) - u_{exp}^{k} \right| + \left| w_{mod}^{k}(E_{sl}, E_{wl}) - w_{exp}^{k} \right|, \qquad \textbf{1}$$

where the sum is over all *DIC subsets* (N, measurement points) in the slab.

Here, we followed the same approach, but used a more recent model which accounts for slab layering and bending-extension
coupling through the weak layer. Applying the concepts of mechanics of layered composites (Jones, 1999), the constitutive
equations of a deforming beam are given by


$$\begin{pmatrix} N \\ M \end{pmatrix} = \begin{pmatrix} A_{11} & B_{11} \\ B_{11} & D_{11} \end{pmatrix} \begin{pmatrix} u_0'(x,z) \\ \Psi'(x) \end{pmatrix},$$ 2

$$Q = kA_{55}(w_0'(x) + \Psi(x)),$$ 3

and link the stiffness quantities of the layered slab, $A_{11}$, $B_{11}$, $D_{11}$ and $kA_{55}$ to the section forces of a free body cut of a differential element of the layered slab. Here $N$ is the normal force, $Q$ the shear force and $M$ the bending moment. The shear correction parameter $k$ is 5/6. The stiffness quantities are the extensional stiffness $A_{11}$, the bending stiffness $D_{11}$, the shear

stiffness $kA_{55}$ and the coupling stiffness $B_{11}$ which incorporates the bending-extension coupling of an asymmetrically layered slab. These stiffness quantities are obtained using the Youngs modulus $E_i$, the shear modulus $G_i$, the Poisson ratio $v_i$, the thickness $h_i$ and the vertical position in the slab $z_i$ for each layer (subscript i).

Data from a manual snow profile provide the thickness and position of each layer, yet $E_i$, $G_i$ and $v_i$ remain unknown. For a typical snow profile, this could easily result in 10 to 50 unknowns. We therefore reduced the degrees of freedom by regarding

each layer as isotropic ($G_i = E_i/2(1+ v_i)$) and setting the Poisson ratio to a constant value $v_i = v = 0.2$. Since the elastic modulus strongly correlates with snow density (Shapiro et al., 1997) we used the manually measured densities of each layer $\rho_i$ to derive $E_i$ using the parametrization suggested by Gerling et al. (2017):

$$E_i = C_0 \left(\frac{\rho_i}{\rho_{ice}}\right)^{C_1}.$$ 4

Here, we fixed the factor $C_0$ to the literature value of 25 540 MPa (Eq. 6; Gerling et al., 2017) and kept $C_1$ as the free nondimensional fitting parameter to minimize the residual ε (Equation 1) and estimate the ideal set of input parameters $C_1$ and

$E_{wl}$. Hence, we accounted for slab layering while limiting the number of degrees of freedom in the optimization. In the following we will call the layered slab model the *LS model*.

For comparison, we also treated the entire slab as homogeneous and isotropic. Then the Young modulus $E$ and Poisson's ratio $v$ of the slab determine the laminate stiffness quantities:

$$A_{11} = \frac{E_{sl}h}{1 - v^2},$$ 5

$$D_{11} = \frac{E_{sl}h^3}{12(1 - v^2)},$$ 6

$$kA_{55} = \frac{5}{6}\frac{E_{sl} h}{2(1 + v)},$$ 7

and $B_{11} = 0$ vanishes. We again used a fixed value for the Poisson ratio of $v = 0.25$. For the uniform slab, Equation 1 is used

again to estimate the slab and weak-layer Young's moduli $E_{sl}$ and $E_{wl}$, respectively. The model assuming a uniform slab we will call *HS model*.

With the elastic modulus of the weak layer and the stress field for the critical cut length $r_c$ determined, the weak-layer fracture energy $w_f$ was obtained by:

$$w_f = G_I + G_{II}, \qquad \text{with: } G_I = \frac{2t}{E_{wl}}\sigma_{wl}(r = r_c)^2 \quad \text{and} \quad G_{II} = \frac{2t(v_{wl}-1)}{E_{wl}}\tau_{wl}(r = r_c)^2,$$ 8





where $G_I$ and $G_{II}$ are the contributions from mode I and mode II, respectively. $\sigma_{wl}(r = r_c)$ and $\tau_{wl}(r = r_c)$ are the
compressive and shear stress in the weak layer at the crack tip, respectively.

We directly followed the methodology of Bergfeld et al. (2021) but used the above, more recent mechanical model which
allows for slab layering and also overcomes a limitation of the model applied by Bergfeld et al. (2021). Their model
conceptualized the weak layer as a set of smeared springs which are attached to the midsurface of a homogeneous slab. That,
however, forced the mode II energy release rate $G_{II}$ in our flat field PSTs to be zero as also the horizontal displacement at the
crack tip $u_{mod}(x = r_c, z = z_{wl})$ is zero.

**2.3 Crack speed**

To compute crack speed during propagation, we followed the method suggested by Bergfeld et al. (2021). They applied a
threshold of five times the standard deviation of the noise in $z$-displacement before crack propagation to the $z$-displacement of
the slab to locate the crack tip at each time step (video frame). Having crack tip location and time, crack speed was estimated
as the first derivative of linear fits of crack tip location with time in overlapping beam sections. A detailed description of the
applied methodology, including uncertainty estimates, is given in Bergfeld et al. (2021). Computing the mean crack speed of
a PST was done by manually defining a range which excludes edge effects (Figure 2, length of the black dashed lines in grey
area). Edge effects were mainly found at the sawing and the far end of the column for PSTs resulting in full propagation, or
near the crack arrest point within the column for PSTs resulting in crack arrest.

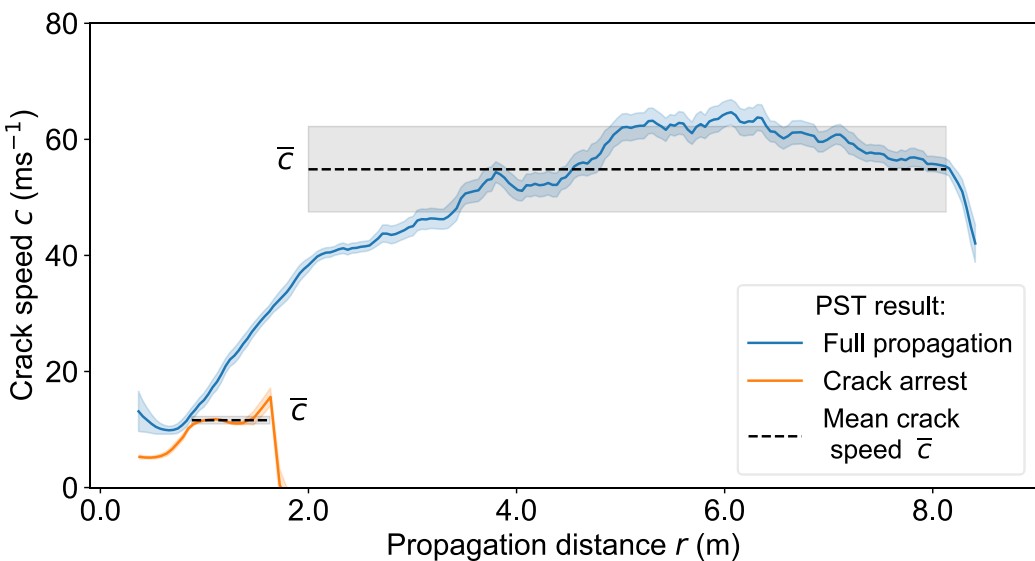

**Figure 2: Crack speed in two exemplary PST experiments: one PST with full propagation (blue), the other with an arresting crack
(orange). The transparent regions indicate the uncertainty. For each PST a range was manually defined in which crack speed was
not influenced by edge effects (length of the black dashed lines in the grey area). Within this range, the mean (value of the dashed
black line) and standard deviation (transparent region) were computed.**





### 2.4 Touchdown distance and crack arrest length

For PSTs resulting in full propagation, the touchdown distance $\lambda$ was defined as the distance between the part of the slab which is still at rest (ahead of the crack tip) and the part which is again at rest after weak layer collapse. To estimate $\lambda$, we followed Bergfeld et al. (2021) by applying a threshold to the downward velocities $w(t)$ of the slab to approximate the distance in which the slab subsides during crack propagation. For each PST resulting in full propagation, the touchdown distance was estimated in every timestep. To derive an average touchdown distance of the experiment, we computed the arithmetic mean and used the

corresponding standard deviation as the uncertainty.

For PSTs resulting in crack arrest, the length of crack propagation was determined by applying a threshold (= the standard deviation of vertical displacements along the PST beam at $t = 0$) to the final vertical displacement. We also used a threshold three times larger than the initial one, and the difference to the initial estimate served as uncertainty.

### 2.5 Crack propagation model

The speed and touchdown distance of propagating cracks in flat terrain were analytically described by Heierli (2005). In principle, Heierli (2005) followed Johnson et al. (2004), who measured a crack propagation speed in flat terrain and described their measurements as a flexural wave in the slab. Hence, Heierli (2005) modelled weak layer cracking as a localized disturbance zone (touchdown distance) propagating as a flexural wave with constant speed $c_{fw}$ and touchdown distance $\lambda_{fw}$:

$$c_{fw} = \sqrt[4]{\frac{g}{2w_{max}}\frac{D_{11}}{\rho h}},$$

$$\lambda_{fw} = \gamma \sqrt[4]{\frac{2w_{max}}{g}\frac{D_{11}}{\rho h}},$$

with $g$ the gravitational acceleration, $w_{max}$ the collapse height, $D_{11}$ is the flexural stiffness (see Equation 6), $\rho$ the mean slab

density, $h$ the slab thickness, $\nu$ the Poisson ratio ($0.25 \pm 0.05$) and $\gamma \approx 2.331$ a factor between touchdown distance and wavelength of the "solitary fracture wave" (Eq. 7b in Heierli, 2005). Uncertainties for modelled crack speed and touchdown distances stem from uncertainties of the required snowpack parameters which we propagated through equations 9 and 10.

### 2.6 Dissipation of dynamic fracture and compaction

The process of a propagating closing crack (also called *anticrack*) can be seen as illustrated in Figure 3. Ahead of the crack

tip, in undisturbed parts of the slab (region I in Figure 3) a beam section (exemplified in Figure 3) is fully supported by the weak layer. The slab – weak-layer system is in a static configuration (Figure 3 and Figure 4a, region I). From a microstructural perspective, the weak-layer support comes from "load chains", e.g. single ice structures carrying small portions of the overall static load of the beam section. As the crack tip approaches, load chains in the weak layer, below a beam section, consecutively fail. The beam section starts to displace downward (Figure 3 and Figure 4a, region II). Subsequently, the advancing downward

movement further breaks weak-layer bonds, but with closer packing it also builds up an increasing number of new bonds in




the weak layer (Figure 3 and Figure 4a, region III). Hence, during this compaction phase the supporting force of the weak layer increases and brings the beam section back to rest (Figure 3 and Figure 4a, region IV).

Comparing this schematic process of a closing crack with an opening crack (e.g. normal mode I crack), the energy needed to fracture the weak layer in region II seems to be analogous to the specific dynamic fracture energy of opening cracks. For

opening cracks, energy dissipation in the weak layer originates only from crack growth and associated physical processes such as surface creation and localized plastic deformation. A closing crack, however, dissipates additional energy behind the crack tip due to secondary fractures and friction during the compaction phase (region III). That is, there are two sources of energy dissipation. The latter we call the *dissipation of compaction*. The energy required to form the crack in the fracture process zone (region II) as the *dissipation of dynamic fracture*.

Assuming that the slab and substratum are in the same stress state before and after crack propagation, slab and substratum contain the same amount of strain energy. Hence, it is reasonable to assume that the total energy dissipated in the weak layer (dissipation of dynamic fracture and compaction) equals the released gravitational potential energy of the slab plus the elastic potential energy of the weak layer.

To separate the dissipation of dynamic fracture from the compaction part, we considered small beam sections within the slab

of the PST (Figure 3, grey area in slab). Every beam section is regarded as behaving like a free body, not attached in the structural compound.

As the crack tip separates the fracture process zone and the compaction zone, an estimate of the location of the crack tip is required. Unlike the crack tip definition in section 2.3, we now follow the schematic process of crack propagation and define the crack tip as the point where all initial load chains failed and there is virtually no support yet of new contacts, it is the point

where the downward acceleration of the beam section is highest. It was confirmed by numerical simulations that this crack tip definition is equivalent to other definitions of crack tip (see Appendix C).

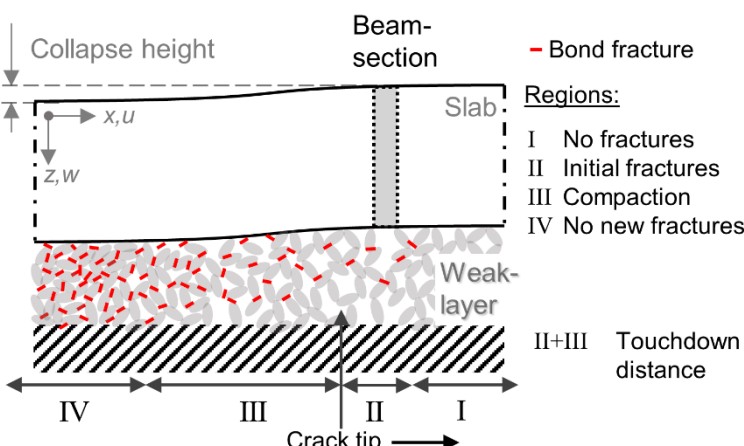

**Figure 3: Schematic representation of a running crack in our flat field PST experiments. The crack tip propagates from left to right. In region I, weak-layer bonds are not yet fractured, while in region IV all weak-layer bonds are broken (red lines). Region II is the**



**fracture process zone, extending from the first bond fractures to the crack tip. In region III, the slab further subsides causing the weak-layer structure to fracture multiple times before closer packing of the weak layer is achieved and the slab comes to rest again.**

We define the time $t_0$ where the peak in the acceleration is reached as the time the crack tip is at the beam section. Everything before $t_0$ is attributed to the initial fracture (dissipation of dynamic fracture), while everything after $t_0$ is part of the compaction phase (dissipation of compaction).

Considering the displacement of the beam section, initially it is at rest with zero displacement (Figure 4a, $t \leq 205$ ms). The supporting force $F_s$ is then equal to the gravitational force $F_g$, induced by the weight of the beam section. Shortly before the crack tip reaches the beam section (Figure 4b, region II), the supporting force $F_s$ decreases and equals the difference between the gravitational force and the acceleration force $F_a$ of the beam section:

$$F_s = F_g - F_a = m \, (g - \ddot{w}),$$ **11**

where $m$ is the mass of the beam section. For $t > t_0$, in region III, the weak layer is compacted and slab support increases. When 250 $F_a/F_g > 1$ (Figure 4b), the slab decelerates before coming to rest again for $t > 310$ ms (Figure 4a).

In each timestep $\Delta t$, the beam section displaces by $\Delta w$. This means that during $\Delta t$ the work $\Delta E^{wl}$ done to destroy the weak layer along $\Delta w$ can be computed as:

$$\frac{\Delta W^{wl}}{\Delta t} = F_s \, \frac{\Delta w}{\Delta t},$$ **12**

$$W^{wl}(t) = \sum_t \Delta W^{wl},$$ **13**

Summing up the increments $\Delta W^{wl}$ provides the total work $W^{wl}(t)$ a beam section did to fracture and compact the weak layer (Figure 4d).

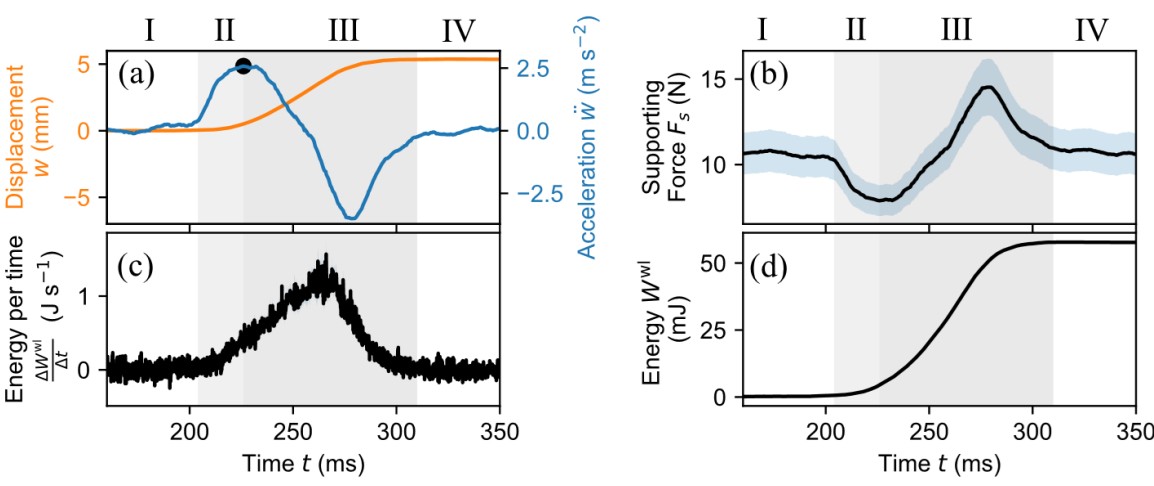


**Figure 4: (a) Vertical displacement $w$ (orange) and acceleration (blue) with time for a beam section in a PST experiment. (b) Supporting force $F_s$ with time. (c) The power $\frac{\Delta W^{wl}}{\Delta t}$ which destroys the weak layer as the beam section displaces. (d) The total work**



Natural Hazards
and Earth System
**$W^{wl}(t)$ done to fracture (region II) and compact (region III) the weak layer. The grey shaded backgrounds separate the regions I to IV defined in Figure 3.**


Separating the work done in region II and III, provides the work done to initially fracture the weak layer in the fracture process zone $W^{frac}(x)$ from the work done to subsequently compact the weak layer $W^{comp}(x)$. Both depend on the $x$-location of the beam section and on the width $b$ and length $l$ of the beam section, where $A = b\,l$ is the area of the beam section which is in contact with the weak layer. Therefore, we define a specific dissipation of dynamic fracture $w_f^{dyn} = W^{frac}(x)/A$, and a

specific dissipation of compaction as $w_{comp} = W^{comp}(x)/A$ (Figure 5).

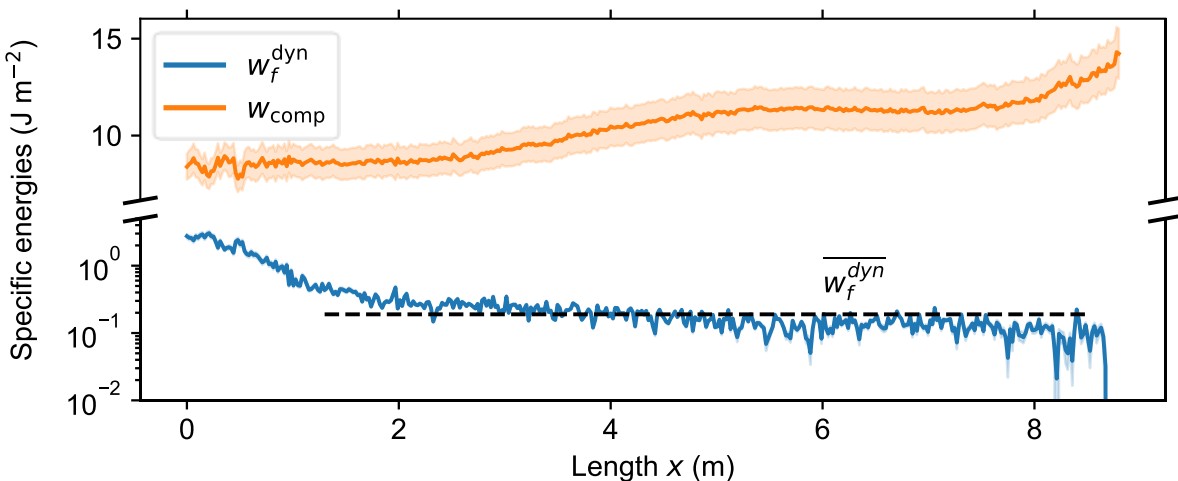

**Figure 5: Specific dissipation of dynamic fracture (blue) and specific dissipation of compaction (orange) over the entire crack propagation length in a PST experiment.**

To neglect edge effects from both ends of the PST column, we manually picked a distance along the column where the specific dissipation of dynamic fracture was almost constant (Figure 5, black dashed line) and computed the mean specific dissipation

of dynamic fracture $\overline{w_f^{dyn}}$ for this distance. Uncertainties in displacement and acceleration of beam sections prior to crack propagation were estimated as the standard deviation. The uncertainties were propagated through Equations. 11 to 13 using Gaussian error propagation.

## 3 Results

On 4 January 2019, the slab was shallow and soft, and it broke while cutting the weak layer in the PST. During the next five

field days, slab thickness and density increased with consecutive snowfalls, and PST results changed from crack arrest to full propagation. This full propagation period lasted around one month, from mid-January to mid-February. Subsequently, PSTs



resulted in crack arrest again. Within the measurement period, the critical cut length increased from 20 to 90 cm, slab thicknesses ranged between 23 and 109 cm, and mean slab density increased from around 110 to 360 kg m$^{-3}$ (Table 1).





**Table 1: Overview of the 24 propagation saw tests (PST) performed between 4 January and 19 March 2019. For each test the propagation result, the critical cut length, the PST column length, slab thickness and mean slab density are given. Values in brackets indicate uncertainties.**

| Date | # PST at day | Test result | Critical cut length (m) | Column length (m) | Slab thickness (cm) | Mean slab density (kg m⁻³) |
|---|---|---|---|---|---|---|
| 4 Jan 2019 | 1 | slab fracture | 0.205 (±0.02) | 2.3 (±0.1) | 23 (±1) | 138 (±7) |
| 7 Jan 2019 | 3 | crack arrest | 0.23 (±0.02) | 3.0 (±0.1) | 46 (±1) | 111 (±6) |
| 8 Jan 2019 | 1 | crack arrest | 0.28 (±0.02) | 3.2 (±0.1) | 42 (±1) | 127 (±6) |
| 9 Jan 2019 | 1 | full propagation[1] | 0.13 (±0.02) | 3.17 (±0.1) | 57 (±2) | 126 (±6) |
| 10 Jan 2019 | 1 | crack arrest | 0.22 (±0.02) | 3.3 (±0.1) | 56 (±2) | 136 (±7) |
| 11 Jan 2019 | 1 | crack arrest | 0.30 (±0.02) | 3.3 (±0.1) | 53 (±2) | 145 (±7) |
| 11 Jan 2019 | 2 | crack arrest | 0.265 (±0.02) | 3.3 (±0.1) | 53 (±2) | 145 (±7) |
| 13 Jan 2019 | 1 | full propagation | 0.33 (±0.02) | 3.3 (±0.1) | 74 (±2) | 148 (±7) |
| 14 Jan 2019 | 1 | full propagation | 0.325 (±0.02) | 4.3 (±0.1) | 107 (±3) | 154 (±8) |
| 15 Jan 2019 | 1 | full propagation | 0.38 (±0.02) | 5.35 (±0.1) | 109 (±3) | 159 (±8) |
| 16 Jan 2019 | 1 | full propagation | 0.395 (±0.02) | 5.3 (±0.1) | 102 (±3) | 187 (±9) |
| 16 Jan 2019 | 2 | full propagation | 0.37 (±0.02) | 7.5 (±0.1) | 102 (±3) | 187 (±9) |
| 17 Jan 2019 | 1 | full propagation | 0.41 (±0.02) | 8.9 (±0.1) | 93 (±3) | 187 (±9) |
| 23 Jan 2019 | 2 | full propagation | 0.57 (±0.02) | 9.0 (±0.1) | 81 (±2) | 194 (±10) |
| 25 Jan 2019 | 1 | crack arrest | 0.63 (±0.02) | 8.8 (±0.1) | 82 (±2) | 217 (±11) |
| 30 Jan 2019 | 1 | full propagation | 0.64 (±0.02) | 9.0 (±0.1) | 89 (±3) | 227 (±11) |
| 2 Feb 2019 | 1 | full propagation | 0.59 (±0.02) | 8.6 (±0.1) | 88 (±3) | 216 (±11) |
| 8 Feb 2019 | 1 | full propagation | 0.64 (±0.02) | 8.5 (±0.1) | 94 (±3) | 247 (±12) |
| 12 Feb 2019 | 1 | full propagation | 0.58 (±0.02) | 8.7 (±0.1) | 107 (±3) | 242 (±12) |
| 18 Feb 2019 | 1 | crack arrest | 0.75 (±0.02) | 8.5 (±0.1) | 86 (±3) | 247 (±12) |
| 22 Feb 2019 | 1 | full propagation[2] | 0.83 (±0.02) | 8.5 (±0.1) | 89 (±3) | 280 (±14) |
| 5 Mar 2019 | 1 | crack arrest | 0.91 (±0.02) | 8.1 (±0.1) | 66 (±2) | 289 (±14) |
| 13 Mar 2019 | 1 | crack arrest | 0.92 (±0.02) | 7.7 (±0.1) | 73 (±2) | 309 (±15) |
| 19 Mar 2019 | 1 | crack arrest | 0.91 (±0.02) | 7.9 (±0.1) | 80 (±2) | 363 (±18) |

[1] From the video analysis the test result "full propagation" can be attributed to an insufficient column length.

[2] The PST was excavated in a way that the column width at the height of the weak layer was only about 20 cm (at a x-location of 4 m). The analysis of the video revealed that the crack was about to arrest before it started to propagate again at about x = 4 m. We therefore suspect the propagation would have resulted in "arrest" with a more precise cutting of the PST column.

## 3.1 Elastic modulus and specific fracture energy

Using a beam bending model (see section 2.2) we estimated the slab elastic properties and weak-layer specific fracture energy
in our PST experiments. We used two models, one with a homogeneous slab (HS model) represented with an effective elastic
modulus and one that accounted for slab layering (LS model) according to the manual density profile.

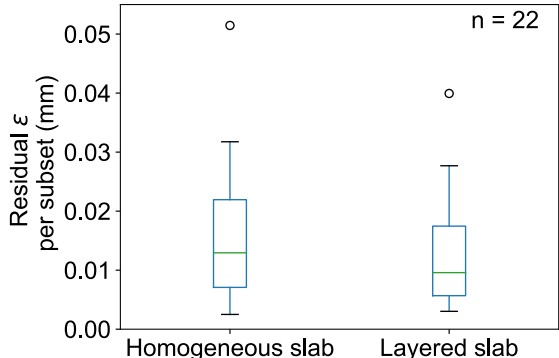

**Figure 6: Distribution of the residuals between the modelled and measured displacement field at the critical cut. The residual is
given as an average over all grid points (DIC subsets). Hence, the residual ε from Equation 1 was divided by the corresponding
number of grid points. The green line indicates the median and the box the interquartile range (first to third quartile).**

Overall, modelled and measured displacement fields matched better when the slab layering was accounted for, although the
differences in residuals were not significant (Figure 6). In the LS model, flexural and tensional stiffness of the slab are coupled.
In our flat field experiments, however, tensile slab loading (horizontal deformation) is much smaller compared to the vertical
deformations induced by slab bending. Hence, the optimization of slab elastic properties is dominated by the vertical
displacements. For the HS model, we thus mostly estimated a flexural elastic modulus rather than a Young modulus. Most
modulus–density parametrizations from literature stem from experiments with different loading conditions. Since in our
experiments the deformation is mostly flexural, the estimated flexural-like elastic modulus is, most likely, a better estimate to
describe the slab behavior than a density parametrization from literature. In the end, the effect of layering, coming along with
tension–bending coupling, gains importance for slope experiments, which are not scope of the present work. Furthermore,
using the different stiffnesses and coupling parameters of the LS model to estimate crack propagation characteristics (e.g. crack
speed, touchdown distance) is not viable since a crack propagation model incorporating a layered slab does not exists yet.
Nevertheless, the results of the LS model (stiffnesses and coupling parameters) are provided in Appendix B.
Using the HS model, in January the effective elastic modulus of the slab $E_{sl}$ rapidly increased by a factor of 20, from 0.5 MPa
to 10 MPa. Thereafter, it was rather constant in February, and slightly increased towards the end of our measurement period
in March (Figure 7a, blue dots). Increases in $E_{sl}$ corresponded to increases in slab thickness $h$ during this period (Figure 7a,
grey line).

During the consecutive snowfalls in the beginning of January, the elastic modulus of the weak layer $E_{wl}$ rapidly increased by

a factor of 10, from around 50 kPa to 0.5 MPa. Afterwards, the increase was less pronounced and peaked in March at around

1 MPa (Figure 7b, dots). Using the LS model, the obtained $E_{wl}$ values were generally about 30 % lower (compare brown and

blue dots in (Figure 7b).

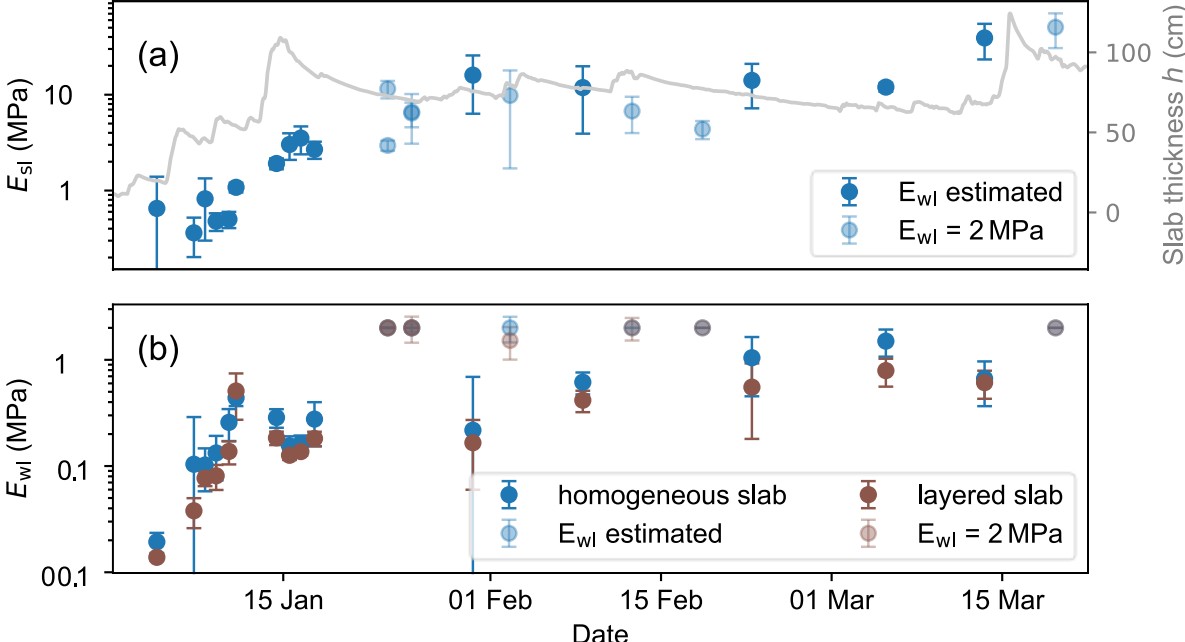

Figure 7: (a) Evolution of effective elastic modulus of the slab $E_{sl}$ (blue dots) and slab thickness (grey line) with time. (b) Elastic
modulus of the weak layer $E_{wl}$ (dots) with time. Blue dots show results for the homogeneous slab model and brown dots those for the
315    layered slab model. Error bars indicate the measurement uncertainty. Solid dots indicate that the elastic modulus of the weak layer
$E_{wl}$ did converge within the boundaries given in the optimization routine. Hence, $E_{wl}$ was estimated. If $E_{wl}$ did not converge it was
kept at the upper boundary of 2 MPa.



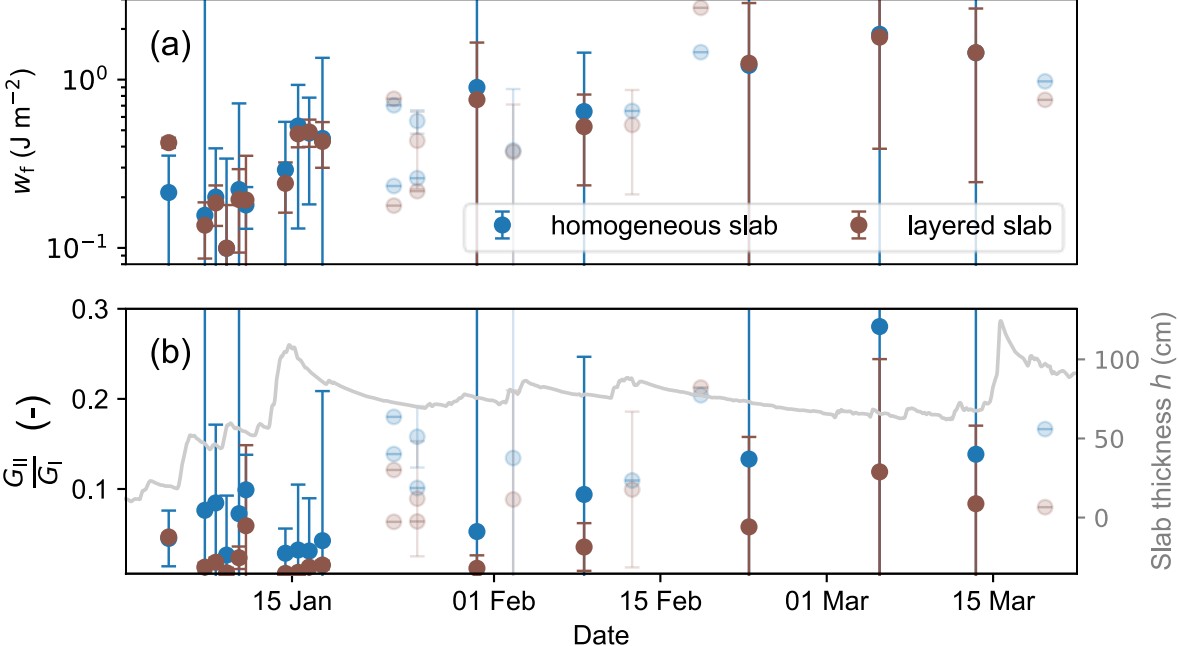

**Figure 8: (a) Evolution of weak-layer fracture energy $w_f$ with time. In general, the layered slab model (brown dots) provided slightly lower estimates of fracture energy compared to the homogeneous slab model (blue dots). (b) Ratio of mode I to mode II energy release rates with time. Compared to the layered model (brown dots), the homogeneous slab model (blue dots) attributed more fracture energy to the shear mode ($G_{II}$), resulting in higher ratios ($G_{II}/G_I$). The layered slab model (brown dots) was less sensitive to changes in slab thickness (grey line). Again, the transparent dots correspond to experiments in which the elastic modulus of the weak layer $E_{wl}$ did not converge within the boundaries given in the optimization routine, and error bars indicate uncertainty.**

Weak-layer specific fracture energies $w_f$ estimated with both models were in the same range (between 0.1 and 1.5 J m$^{-2}$) and generally exhibited an increasing trend with time (Figure 8a). The initial increase was stronger. On average, the increase was 0.02 J m$^{-2}$ per day. Overall, the LS model predicted 22 % lower $w_f$ values than the HS model. Larger differences between both models were observed in the mode composition of the fracture energy (Figure 8b). Throughout the measurement series, the ratio between the shear ($G_{II}$) and the compressive ($G_I$) component of the fracture energy ($G_{II}/G_I$) was substantially lower with the LS model, on average 51 %.

### 3.2 Dissipation of dynamic fracture and compaction

On average, the specific dissipation of compaction was 30 times higher than the specific dissipation of dynamic fracture, indicating that the majority of the energy is used for weak-layer crushing and not for advancing the crack in the weak layer (Figure 9a and b, respectively). The specific dissipation of dynamic fracture generally increased with time. On 4 January it was $5 \pm 16 \times 10^{-3}$ J m$^{-2}$ and at the end of the measurement series on 19 March it was $0.43 \pm 0.19$ J m$^{-2}$ (Figure 9a). These values were in the same range as the static weak-layer specific fracture energy $w_f$ (Figure 8a). Specific dissipation of dynamic fracture did not depend on the outcome of the PST as values from full propagation and crack arrest aligned well in the temporal trend.




The specific dissipation of compaction, on the other hand, was significantly lower for PSTs resulting in crack arrest than for PSTs resulting in full propagation (Figure 9b), on average $2 \pm 0.3$ J m$^{-2}$ and $10.1 \pm 0.4$ J m$^{-2}$, respectively.

As the dynamic fracture in weak snow layers is always accompanied by a reduction in the volume of the weak layer, a volumetric fracture energy might be more appropriate than the classical energy per area. As alternative to the specific dissipation of dynamic fracture, expressed as an energy per fractured area, we therefore computed a volumetric fracture dissipation expressed as the energy needed to compact the weak layer (Figure 9c). To do so, we divided the specific dissipation of dynamic fracture by the settlement of the slab at the crack tip (see Figure 4):

$\overline{w_f^{vol.\ dyn}} = \overline{w_f^{dyn}} \Big/ w(t(\max(\ddot{w}))$. Overall, the volumetric dissipation of dynamic fracture increased with time from

$1 \pm 4$ kJ m$^{-3}$ to $2.7 \pm 0.6$ kJ m$^{-3}$.



**Figure 9: (a) Specific dissipation of dynamic fracture $\overline{w_f^{dyn}}$ with time. (b) Specific dissipation of compaction $\overline{w_{comp}}$ with time. (c) Volumetric dissipation of dynamic fracture $\overline{w_f^{vol.\ dyn}}$ with time. PST results are shown with different colors: full propagation (blue dots) or crack arrest (orange dots). Error bars indicate the measurement uncertainty.**

### 3.3 Crack speed

Crack speed increased during the first nine field days until the highest crack speeds were observed on 16 and 17 January ($55 \pm 8$ m s$^{-1}$ and $55 \pm 7$ m s$^{-1}$, respectively). Thereafter, crack speed decreased and remained almost constant around $36 \pm 4$ m s$^{-1}$ (Figure 10a). In addition to the measurements obtained from the PSTs, crack speeds were also computed from snowpack parameters (Figure 10a, grey dots) using Eqs. 9 and 6. For the slab elastic modulus the estimates of the HS model

were taken. The modelled speeds were in good agreement, especially during the first five field days and after the end of January

(average deviation 7 m s⁻¹). The greatest mismatch was for the days when crack speeds were largest.

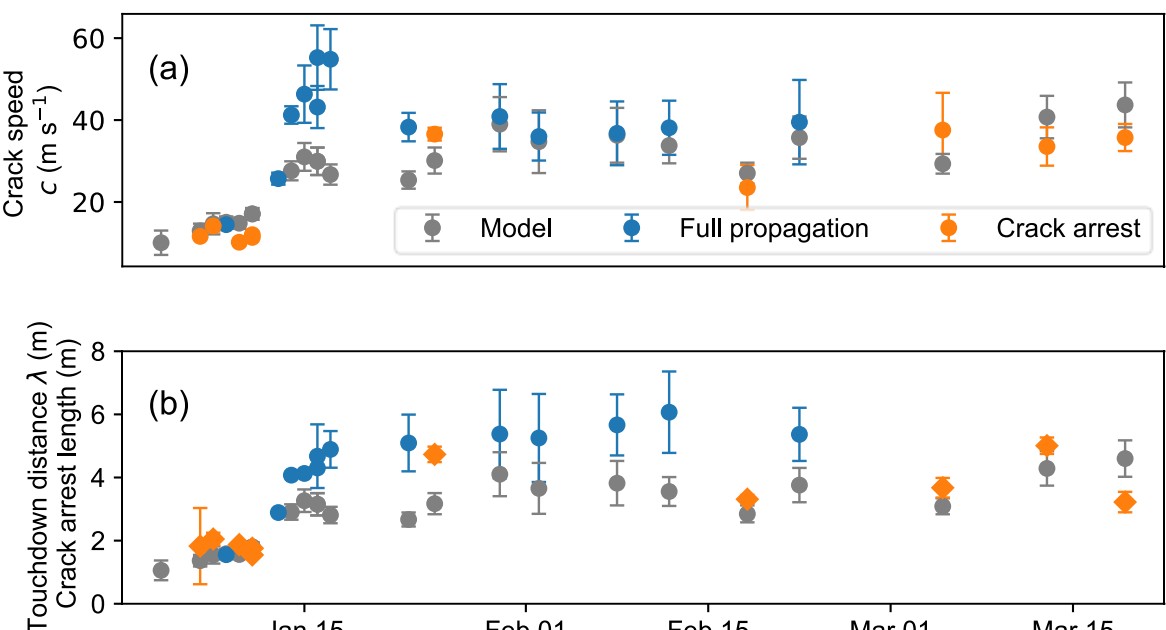

**Figure 10: (a) Crack propagation speed with time. (b) Touchdown distance and crack arrest distance with time. PST resulting in full propagation are shown with blue dots, crack arrest with orange dots. Grey dots indicate (a) modelled crack speed and (b) touchdown distance obtained with snowpack parameters using equations 9 and 10. For PSTs resulting in crack arrest, the crack**
**arrest length was estimated (orange diamonds). Vertical bars indicate the measurement uncertainty.**

### 3.4 Touchdown distance and crack arrest length

For PSTs resulting in full propagation, the touchdown distance can be estimated, given that the column length was larger than the touchdown distance (Figure 10b, blue). In total, we measured touchdown distances in 13 PSTs from mid-January to the end of February. In this period, the touchdown distance increased from $1.55 \pm 0.14$ m to $6.1 \pm 1.3$ m. The overall trend is well
reproduced with the model (Equation 10, grey dots in Figure 10b), although on average the modelled touchdown distances were 38 % lower. For PSTs resulting in crack arrest, the distance to the point of crack arrest was estimated. With the exception of 9 January, crack arrest length was consistently shorter than the touchdown distance measured prior to or following the crack arrest experiment.



## 4 Discussion

### 4.1 Elastic modulus and specific fracture energy

For all 24 PST experiments, the deformation field of the slab during sawing was derived via image correlation analysis of individual movie frames. The deformation field and the corresponding cut length (location of the snow saw) served as input to estimate the elastic properties of the slab and the specific fracture energy of the weak layer. For that, two mechanical models predicting slab deformation were used. In the HS model the slab was represented as homogeneous with mean density and effective elastic modulus. The LS model accounted for slab layering.

The effective elastic modulus estimated with the HS model, showed a rather steady increase with time. The consecutive large snowfalls were not individually reflected in the measurement of the effective modulus. But these rapid increases of slab thickness strongly changed the layering and therefore the bending behavior of the slab (Figure 7a). The stiffness properties obtained with the LS model showed both an overall increase in stiffness, resulting from densification, sintering and loading, as well as a close relation to the individual snowfall events influencing the layering (Appendix B). Therefore, we deem the LS model to be more precise, even though the LS model did not show a significantly lower residual between measured and modelled deformation field (Figure 6).

In fact, many studies employing the finite element method have shown that slab layering is important for failure initiation as well as the onset of crack propagation (Habermann et al., 2008;Monti et al., 2016;Schweizer et al., 2011;van Herwijnen et al., 2016). In contrast, a homogeneous slab model coming along with an "equivalent modulus" is not able to represent important aspects of the deformation behavior of a layered slab (e.g. extension-bending coupling, vertical shift of zero bending line). In general, if the stiffness tensor is expressed by two engineering constants (e.g. equivalent modulus and Poisson's ratio) to relate stress and strain, a homogeneous material (not layered and isotropic) is assumed. A layered snow slab does not match this condition and the corresponding stiffness tensor cannot be expressed with two independent variables and more parameters are needed. Hence, an "equivalent modulus" may incorporate viscosity and symmetrical layering (around the middle of the slab), but since it is a single elastic property, only if the deformation field is dominated by a single eigenmode such as bending. If more eigenmodes are relevant or the slab is asymmetrically layered, an "equivalent modulus" cannot accurately model the complex deformation field. However, most natural snowpacks are asymmetrically layered and deformation is in general multimodal.

The result that our LS model did not show a significantly lower residual can be attributed to our flat field experiments where the self-induced loading of the slab is perpendicular to the layering. The eigenmode of slab deformation is mainly flexural bending and downslope pull from the unsupported part of the PST or lateral shear deformation are negligible. The absence of downslope pull induced tension makes the tensional stiffness and the extension-bending term less impactful in flat field PSTs, severely limiting the benefits of the LS model.

Regarding the elastic modulus of the weak layer, both the HS and LS model did not converge for the same six experiments (Figure 7b, transparent dots). For the remaining experiments, both models estimated similar trends with time. The consistently





lower elastic modulus estimates from the LS model can be attributed to the limitations of the HS model. Indeed, for most natural slabs density and stiffness increase with depth, as was also the case in our measurement series. Due to this asymmetry, the zero-bending line (neutral axis) in the slab shifts downwards, resulting in less horizontal displacement at the bottom of the

slab close to the weak layer. This effect cannot adequately be reproduced by the HS model, resulting in a higher weak-layer elastic modulus to account for the small measured horizontal displacement at the bottom of the slab.

This shortcoming of the HS model is also seen in estimates of the weak-layer specific fracture energy and especially its mode decomposition $G_{II}/G_I$ (Figure 8). On average, the LS model estimated a 22 % lower fracture energy. Regarding the mode contribution, the differences are larger. Again, the horizontal displacements at the crack tip in the HS model are inherently too

large. This directly affects the specific fracture energy estimates using Equation 8. The mode II contribution of specific fracture energy is therefore also too high. This can be nicely seen by comparing the first and second experiment. In the first measurement, the slab consisted of a single 23 cm thick layer, hence, it was rather homogeneous. The HS and LS model predicted the same mode contribution (Figure 8b, first data point). Three days later, an additional 35 cm of new snow was added to the slab. The slab was more layered and the mode contributions were completely different from the two models.

Subsequently, slab layering changed, but did not vanish, hence differences between both models persisted until the end of the measurement series. Kirchner et al. (2002) and Schweizer et al. (2004) were the first considering the mode contributions in fracture toughness measurements. They used notched cantilever-beams consisting of snow types which typically form the slab layer (e.g. rounded grains). Up to now, the mode composition of fracture energy for a weak snow layer is not considered in experimental data. Only Bergfeld et al. (2021) discussed the contributions, but they were not able to estimate a mode II

contribution since the underlying mechanical model (Rosendahl and Weissgraeber, 2020a) did not allow horizontal displacement at the crack tip for experiments without slope angle. Thus, our measurements are the first to differentiate the components of fracture energy, which is necessary to better understand weak-layer resistance to cracks under different loading conditions (Rosendahl and Weissgraeber, 2020b). Self-induced loading in a PST by the unsupported part of the slab always induces a combination of compressive and shear stresses at the crack tip, even in flat field tests (Gaume et al., 2018a). Hence

PSTs are always mixed-mode fracture tests and the mode contributions have to be indicated. A weak layer thus exhibits different apparent fracture energies depending on whether it is tested at 0° or 35° slope angle. Similar, the bending behavior of slabs with different layering resting on the same weak layer can induce different loading conditions at the crack tip, resulting in apparently different fracture energies of the weak layer. Hence, without considering the mode contributions, the derived fracture energies are difficult to interpret and their use in other studies is hampered. A more comprehensive understanding of

crack propagation in weak layers requires fracture energy measurements covering a wide range of loading conditions in mixed-mode crack propagation, from closing mode I (compression) to mode II (shear). An improved understanding of the fracture envelope of weak layers is a prerequisite for reliably modelling crack propagation at different slope angles.

Regarding the temporal evolution of weak-layer specific fracture energy, it increased with time as did the elastic modulus of the slab and weak layer. From 4 to 20 January fracture energy increased by around one order of magnitude, from 0.1 to

1.5 J m$^{-2}$, a similar increase as was observed by Schweizer et al. (2016).





Previous studies (Gaume et al., 2014;Birkeland et al., 2019) assumed a power-law increase of weak-layer fracture energy with time, as is typical for snow sintering (e.g. van Herwijnen and Miller, 2013). Our measurements suggest a linear trend of around 0.02 J m$^{-2}$ d$^{-1}$ for the buried surface hoar layer we tested. Jamieson and Schweizer (2000) investigated changes in shear strength of buried surface hoar layers (mean: 56 Pa d$^{-1}$) and suggested that the strengthening was due to the penetration of the surface

hoar crystals into adjacent layers, rather than to sintering. A finding that applies specifically to weak layers consisting of surface hoar. Most likely, strengthening and the increase in fracture energy are due to the same microstructural changes. Therefore, it can be assumed that the trend of fracture energy with time is different for different grain types. Since there are many types of weak layers, besides buried surface hoar, more data are needed to confirm that our findings are generally applicable.

**4.2 Dissipation of dynamic fracture and compaction**

The specific fracture energy estimated from the onset of crack propagation (static fracture energy) does not necessarily coincide with the required energy during dynamic propagation (Freund, 1990). As the dissipation of dynamic fracture in closing cracks is superimposed with the dissipation of compaction, we suggested to separate the two contributions (Section 2.6). Throughout the measurement series, the dissipation of dynamic fracture was lower than the static fracture energy (compare Figure 8a and

Figure 9a), although the values were in the same range.

The dissipation of dynamic fracture had two local maxima with time (Figure 9), first around 15 January, and second around 18 February. The dissipation of dynamic fracture is not necessarily a material property of the weak layer, as it may also depend on crack propagation characteristics, such as crack speed. The high values of fracture energy around 15 January are well correlated with higher crack speeds (Figure 10a), hence higher propagation speeds may lead to higher dissipation of dynamic

fracture as it was also observed in bones (Behiri and Bonfield, 1980) or some engineering plastics (e.g. Fond and Schirrer, 2001). On the other hand, the second maximum on 18 March was not characterized by a high crack speed and this experiment resulted in crack arrest. We therefore attribute the observed fluctuations in the dissipation of dynamic fracture to the method we used to separate dissipation of dynamic fracture from the compaction part. Analogous to opening cracks, we envisioned and estimated dissipation of dynamic fracture as the energy dissipated in the weak layer ahead of the crack tip (uncracked

region). For opening cracks, this is close to the total energy dissipated in the weak layer, since there is typically no energy dissipation in the weak layer behind the crack tip (cracked region). In the case of closing cracks, more energy is dissipated in the weak layer behind the crack tip to crush the weak layer, what we call dissipation of compaction. This is not only a material property of the weak layer, but a quantity that likely depends on the entire system of slab, substratum, weak layer and slope angle. Our estimates for the dissipation of dynamic fracture are therefore closely related to our definition of the crack tip and

the associated amount of settlement (amount of interpenetration) at the crack tip. We therefore also determined a volumetric dissipation of dynamic fracture by dividing the consumed energy per cracked area (units J m$^{-2}$) by the settlement (or vertical displacement) at the crack tip. This volumetric dissipation of dynamic fracture is the energy per destroyed/compacted volume,



a measure that seems more intuitive for closing cracks. For the volumetric dissipation of dynamic fracture, the earlier mentioned maxima in January and March disappeared, and there was overall less fluctuation with time (Figure 9c).

The computed volumetric dissipation of dynamic fracture only accounts for mode I contributions, as we only considered the vertical displacement. In our flat field experiments, this is likely not problematic as mode II contributions are expected to be small (cf. Figure 8b). For PST experiments on slopes, the applied methodology will therefore have to be adapted to incorporate the mode II contributions in the volumetric dissipation of dynamic fracture.

The dissipation of compaction separates PSTs resulting in crack arrest and full propagation (Figure 9b, orange and blue dots,

respectively). In PSTs with crack arrest no stable crack propagation occurred, even if the crack propagated several meters. The collapse height decreased steadily towards the crack arrest point. Therefore, it is not surprising that the dissipation of compaction in PSTs resulting in crack arrest was also lower as the overall settlement was much lower (Figure 9b). This clearly highlights that the dissipation of compaction as defined in our method is not a material property of the weak layer, and rather a property of the entire system.

**4.3 Crack speed, touchdown distance and crack arrest length**

In our measurement series, crack speed values initially increased, peaked around 16 January 2019 and were then rather constant until the end of the measurement period. The peak in crack speed coincided with days of high avalanche danger level and the observation of several large and very large natural avalanches within a few kilometers of our test site. In general, the measured crack speeds are in the same range as crack speeds recently determined in numerical simulations (Bobillier et al., 2021;Trottet

et al., 2022;Bobillier et al., 2022). Bobillier et al. (2022) and Trottet et al. (2022) further report a possible transition to much higher crack propagation speeds on slopes. However, since the present PST series were made in the flat, this transition is not expected and has not been observed either.

Measured touchdown distances, only obtained for PSTs resulting in full propagation, were consistently higher than predictions of the crack propagation model (Figure 10, grey dots). This discrepancy is in part attributed to an incorrect model assumption

that the slab is in free-fall motion during weak-layer collapse (Bair et al., 2014;Bergfeld et al., 2021). Other contributing factors are uncertainties in the model inputs, namely thickness, load and elastic modulus of the slab, as well as the collapse height (Equation 10). With the exception of the elastic modulus, these parameters can readily be measured. For snow the true (high-frequency, small strain) elastic modus should be distinguished from measurements which are likely affected by the viscoplasticity of ice. While the true elastic modulus can just be determined with acoustic wave propagation measurements or

with micro-computed tomography based finite element calculations (Capelli et al., 2016;Gerling et al., 2017), other measurement techniques and applications act at lower strain rates which incorporate viscous and/or plastic effects. In this regime the elastic modulus is an effective modulus depending on the strain rate. We used the elastic modulus obtained during the sawing phase of the PST. Sawing takes a few seconds during which the slab bends typically less than a millimeter. During crack propagation, the slab undergoes a 10 times larger deformation in around one hundredth of the time of sawing



(around 1 mm in 20 ms). Hence, the strain rate during crack propagation is at least two orders of magnitude higher. A higher elastic modulus, measured at higher strain rates, would thus be a more appropriate model input.

Initially, touchdown distance increased rapidly, as did the flexural rigidity of the slab (cf. Equation 10). After 17 January, touchdown distance values were similar with an average of $5.4 \pm 0.3$ m. Long touchdown distances indicate that it takes a considerable distance before cracks are no longer influenced by the crack initiation and propagate in a self-sustained state.

Basically, two factors contribute to edge effects when crack propagation starts. First, energy release rate is an increasing function with crack length, but just as long as the slab does not yet come to rest again. After one touchdown distance the energy release rate is bounded. Second, the energy release rate is influenced by the triggering mechanism. In a PST, the saw creates a gap in the weak layer. The slab is free hanging above the saw cut and no dissipation of compaction is needed during settlement. Hence, the undercut part of the slab releases more energy during settlement than later on during self-sustained crack

propagation. Similarly, the additional load induced by typical triggers of slab avalanches (e.g. skier or explosive) leads to initially increased energy release rates. This effect fades with crack propagation distances and is negligible after one touchdown distance. Whether a snowpack can then propagate a crack over long distances is only revealed, after at least one touchdown distance, when self-sustained crack propagation is observed. In other words, to get reliable information about sustained crack propagation propensity, field tests must be much longer than the touchdown distance (Bergfeld et al., 2021).

Our measurements revealed crack arrest after propagation distances of more than 4 meters, which is about double the length of normal PST experiments. Crack lengths in PSTs with crack arrest were, however, always shorter than touchdown lengths measured in experiments shortly before or after (Figure 10b, blue and orange). The only exception was the experiment on 9 January 2019 when we observed a crack that almost arrested. In this experiment, the downward speed of the slab almost decreased to zero when the slab at the sawing end of the experiment came to rest on the substrate, i.e. the crack propagated

one touchdown length. This suggests that the energy release rate was no longer sufficient for crack propagation and the crack continued to propagate by consuming the kinetic energy stored in the slab. When the kinetic energy was almost depleted, and all velocities of the slab were close to zero, the crack almost arrested. At this point, however, the crack tip was already influenced by the free end of the PST column. The energy release rate increased again and the crack was able to propagate again for the last 40 cm to the end of the column. In a longer PST experiment, the crack would likely have arrested.

The experiments performed after 17 January had a rather thick slab (~ 1 m) with an elastic modulus around 10 MPa and a collapse height of the weak layer of less than 10 mm. For other snowpack stratigraphies associated with avalanches, such as thick, dense slabs (high elastic modulus) on a thick layer of depth hoar (large collapse height), we expect touchdown distances to be even longer. Observing self-sustained crack propagation in PSTs will then become very unpractical as it will require even longer experiments.

**4.4 Sustained crack propagation index**

Typically, the snowpack does not tend to propagate fractures when the slab is soft with a low tensile strength, or the load is insufficient. As the slab densifies and/or the load increases, the snowpack then tends to propagate a crack in the weak layer,



provided one exists. However, as the slab thickness and density further increase, the slab becomes structurally stiffer, resulting in lower stress concentrations at the crack tip, and a self-propagating crack becomes less likely. Consequently, the probability

of self-sustained cracking is a function of time with one or more local maxima. Our measurement series showed this temporal evolution. During the life cycle of the tested weak layer, PSTs initially resulted in slab fractures, followed by crack arrest, full propagation and then crack arrest again.

If instability has to be assessed from layer properties rather than from PST experiments, the critical crack length is typically the metric used to describe the propensity of a snowpack to propagate a crack along the weak layer (Reuter et al., 2015).

However, in the absence of load increase, parametrizations of the critical crack length result in a steady increase with time (Gaume et al., 2017;Richter et al., 2019). Thus, they cannot fully represent the actual temporal trend of an instability. Moreover, the critical crack length does not fully reflect the result of a PST, since a PST also provides important information about the type of the propagation, i.e. slab fracture, crack arrest or full propagation.

While the critical crack length can easily be estimated from measured or simulated snow profiles (Gaume et al., 2017;Richter

et al., 2019), the outcome of a PST remains unknown. Reuter and Schweizer (2018) suggested a tensile failure criterion to indicate whether a slab fracture is likely to occur. However, our measurement series showed that before and after the full propagation period, the snowpack tended to arrest cracks without slab fractures. There is currently no method to assess crack arrest, and an index for the propensity of self-sustained crack propagation is missing. Our results showed that the highest crack speeds (Figure 10, around 15 January) were observed on days when concurrently avalanche danger was 4-High or 5-Very high

(Figure 11, colors at top) and major avalanche activity (Figure 11, brown stars, data set described in Schweizer et al. (2021)) was observed.

We therefore suggest an index which estimates the propensity of the snowpack to support self-sustained crack propagation ($SSP$) as the ratio of the square of crack speed $c^2$ and the critical cut length $r_c$:

$$SSP = \frac{c^2}{g\,r_c},$$   **1**

where $g$ is the gravitational acceleration owing to make the index dimensionless.

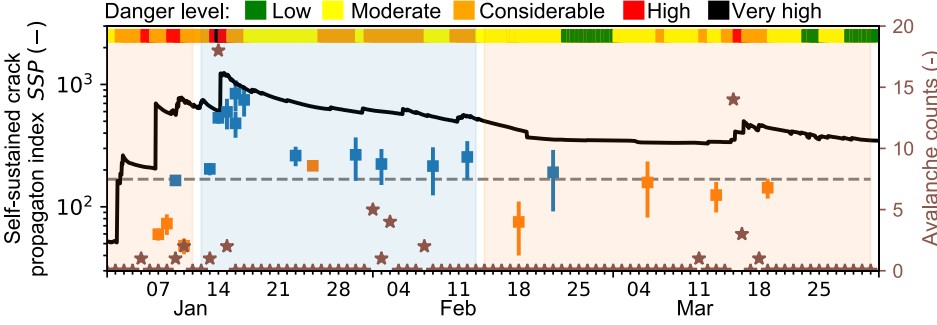


**Figure 11: Index of self-sustained crack propagation SSP (blue and orange squares for full propagation and crack arrest, respectively) with time. For comparison, regional avalanche activity (brown stars), given as total counts of observed avalanches with size 3 or larger in a 10 km radius around our field site, and the regional avalanche danger forecast (colours on top) are given. The**





**blue and orange background indicate the periods when PSTs mainly resulted in crack arrest and full propagation, respectively. The self-sustained crack propagation index SSP was also modelled for simulated snow stratigraphy (black line) using crack speed and critical cut length parametrizations from Heierli (2005) and Richter et al. (2019), respectively.**

We derived the *SSP* index from PST resulting both in crack arrest and full propagation (Figure 11, orange and blue squares respectively). For the period when the experiments showed mainly full propagation (Figure 11, blue background), the *SSP* indices derived from the measurements were larger than in the periods with crack arrest (orange background). The five highest

*SSP* indices (between 14 and 17 January) coincided with high avalanche activity and avalanche danger level 5-Very high (both on the 14 January, Figure 11, brown stars and colors on top of the panel, respectively). To discriminate between the period when the PSTs resulted in full propagation from the periods when the cracks arrested, we suggest a threshold of *SSP* = 168, for our data set (Figure 11, grey dashed line).

In addition, we also derived *SSP* from simulated snow stratigraphy obtained with the numerical snow cover model

SNOWPACK (Appendix D) for our field site with an hourly resolution. The SNOWPACK output was used to compute crack propagation speed (Eq. 11; Heierli, 2005; Fig. E1b, grey line) and critical cut length (Richter et al., 2019; Fig E1b, red line). Elastic properties of the slab were derived from the mean density of the slab using the parameterization suggested by Scapozza (2004). The peak of the model-derived *SSP* (Figure 11, black line) coincided with the maximum values of *SSP* obtained from the measurements (around 15 January). However, discrepancies between the measurement- and model-derived values of *SSP*

are present. The modelled values were generally larger than the ones from the measurements which suggests also a different, larger threshold (around 500). Additionally, the first two local maxima of the modelled values of *SSP* (3 and 6 January) overestimated the propensity for self-sustained crack propagation, as the parameterizations likely do not correctly account for the soft and shallow slab, which was present at those days and did not support self-sustained crack propagation.

Therefore, the *SSP* index represents a first attempt only to estimate the propensity of the snowpack to support self-sustained

crack propagation. While it was in line with the results of our PST series, future studies will have to show whether the index is also useful to estimate avalanche size.

## 5 Conclusions

We conducted a series of 24 flat field PST experiments, up to ten meters long, over a 10 weeks period. All PST experiments were analyzed using digital image correlation to derive high-resolution displacement fields. From the displacements we derived

snow properties (elastic properties, specific fracture energy) and crack propagation metrics (dynamic energy dissipations, crack speed, touchdown distance, crack length).

To estimate the elastic properties of slab and weak layer we used two mechanical models. One considered the slab as homogeneous (HS) and the other accounted for slab layering (LS). From the HS model we derived the effective elastic modulus of slab and weak layer as well as the specific fracture energy of the weak layer. As PST experiments are characterized by

mixed mode loading, we separated the specific fracture energy into mode I and mode II contributions. With the LS model, we estimated the same properties of the weak layer. With both models we obtained specific fracture energies between 0.1 and





1.5 J m$^{-2}$. The behavior of the layered slab was given by stiffness quantities (Appendix A). Comparing both models, the LS model provided displacement fields which were slightly closer to the measured displacements. This superiority would probably be more pronounced for PST experiments on steep slopes. Furthermore, the models differed in the estimates of weak-layer

elastic modulus and the mode contribution of the fracture energies. These differences were attributed to the fact that an effective elastic modulus of the slab cannot represent the asymmetric layering of the slab. As the estimation of the specific fracture energy is based on the displacements at the crack tip, this drawback of an effective elastic modulus also propagated to the estimates of specific fracture energy, in particular in the contributions from mode I and II ($G_{II}/G_I$ was 50 % lower for the LS model).

In addition to the (static) specific fracture energy at the onset of crack propagation, we also computed a dissipation of dynamic fracture during crack propagation by separating the work done in the weak layer into a dissipation of dynamic fracture which is absorbed in the fracture process zone ahead of the crack tip, and the dissipation of compaction absorbed after the crack tip passed, and the slab further bends down and settles. The dissipation of compaction was 30 times higher than the dissipation of dynamic fracture ($5 \times 10^{-3}$ J m$^{-2}$ to 0.4 J m$^{-2}$) which was in the same range as the static specific fracture energy of the weak

layer. As the dissipation of dynamic fracture and compaction is inherently linked to a certain amount of interpenetration of the weak layer, we alternatively computed a volumetric dissipation of dynamic fracture. The volumetric dissipation of dynamic fracture (1 to 2.7 kJ m$^{-3}$) exhibited a steadier increase during the measurement series and was therefore deemed to be a useful quantity to express the resistivity of the weak layer against closing cracks.

Before and after the time period when the PSTs resulted in full propagation, PSTs resulted in crack arrest. Crack arrest lengths

were more than 5 meters and therefore longer than common PSTs column lengths. However, crack arrest lengths were always shorter than touchdown distances, measured in full propagation PSTs conducted before or after the crack arrest experiment. This indicates that self-sustained crack propagation can only be assessed with PSTs, if the column length is larger than at least one touchdown distance, i.e. about 8 m. For shorter column length, crack propagation is affected by the artificial crack initiation and not yet in a self-sustained mode.

Our measurement series not only provided valuable insight in self-sustained crack propagation, but also motivated us to suggest an index describing the propensity of self-sustained crack propagation. The index is based on the ratio of the square of crack speed and critical cut length. For our measurement series, we found large index values for PSTs resulting in full propagation and low values for arresting cracks. As the index could potentially be used to estimate the likelihood of very large avalanches, we compared it to the number of avalanches (larger than size 3, observed within a 10 km radius around the field site) and the

local avalanche danger level. In this comparison, the index derived from the field measurement showed a quantitively good agreement. In contrast, agreement was lower, but still reasonable, when we computed the index from simulated snow stratigraphy. The proposed index may be helpful to estimate avalanche size and thus improve avalanche forecasting once validated with a data set that includes avalanche release size.



The presented data set is valuable for validation of numerical models as we tracked the propagation characteristics of a single weak layer over a ten-week period. Since our data set contains only flat field experiments, future measurements should be conducted on slopes under different loading angles.

*Data availability.* High-speed recordings and processed data will become available at www.envidat.ch.

*Author contributions.* JS and AH designed the research and together with BB and GB developed the experimental setup. BB and AH carried out the experimental work. BB processed the high-speed recordings. PR, PW, VA helped and partly performed the analysis of the mechanical models to assess mechanical properties and JD contributed to all parts of the project. The manuscript was written by BB with input from all authors.

*Competing Interests:* The authors declare that they have no conflict of interest.

*Acknowledgements:* Achille Capelli, Christine Seupel, Colin Lüond, Alexander Hebbe, Simon Caminada, Bettina Richter and Stephanie Mayer assisted with fieldwork. Flavia Mäder helped with the DIC analysis and Frank Techel provided the avalanche activity data.

*Financial support:* BB and GB have been supported by a grant from the Swiss National Science Foundation (200021_169424).

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





**Appendix A: Camera and digital image correlation settings.**

**Table A1: Settings of the high-speed camera Phantom, VEO710. The horizontal image resolution was kept constant at 1280 pixel.**

| Date | # PST at day | Frame rate (fps) | Image height (pixel) | Focal length of lens (mm) | Lens aperture (-) | Number of frames (-) | Pixel conversion (mm/pixel) |
|---|---|---|---|---|---|---|---|
| 01.04.2019 | 1 | 20000 | 256 | 24.0 | 2.8 | 8001 | 2.26 |
| 01.07.2019 | 3 | 14000 | 400 | 39.0 | 3.4 | 8001 | 2.48 |
| 01.08.2019 | 1 | 3000 | 400 | 24.0 | 2.8 | 10000 | 2.57 |
| 01.09.2019 | 1 | 14000 | 400 | 24.0 | 2.8 | 10001 | 2.85 |
| 01.10.2019 | 1 | 14000 | 400 | 24.0 | 2.8 | 9001 | 2.88 |
| 01.11.2019 | 1 | 14000 | 400 | 24.0 | 2.8 | 8001 | 2.98 |
| 01.11.2019 | 2 | 10000 | 400 | 32.0 | 2.8 | 9001 | 2.68 |
| 01.13.2019 | 1 | 7000 | 400 | 24.0 | 2.8 | 8001 | 2.82 |
| 01.14.2019 | 1 | 7000 | 400 | 24.0 | 2.8 | 7001 | 3.64 |
| 01.15.2019 | 1 | 7000 | 352 | 24.0 | 2.8 | 8001 | 4.38 |
| 01.16.2019 | 1 | 7000 | 352 | 24.0 | 2.8 | 6001 | 4.25 |
| 01.16.2019 | 2 | 7000 | 352 | 24.0 | 2.8 | 7001 | 6.33 |
| 01.17.2019 | 1 | 10000 | 256 | 24.0 | 2.8 | 9001 | 6.98 |
| 01.23.2019 | 2 | 7000 | 256 | 24.0 | 2.8 | 6001 | 7.08 |
| 01.25.2019 | 1 | 7000 | 304 | 24.0 | 2.8 | 6001 | 9.07 |
| 01.30.2019 | 1 | 15000 | 304 | 24.0 | 2.8 | 7001 | 7.11 |
| 02.02.2019 | 1 | 7500 | 304 | 24.0 | 2.8 | 7001 | 6.85 |
| 02.08.2019 | 1 | 10000 | 240 | 24.0 | 2.8 | 7001 | 6.92 |
| 02.12.2019 | 1 | 20000 | 240 | 24.0 | 2.8 | 8501 | 6.88 |
| 02.18.2019 | 1 | 18000 | 240 | 24.0 | 2.8 | 6001 | 6.76 |
| 02.22.2019 | 1 | 5000 | 240 | 24.0 | 2.8 | 6001 | 6.8 |
| 03.05.2019 | 1 | 20000 | 240 | 24.0 | 2.8 | 5001 | 8.63 |
| 03.13.2019 | 1 | 22000 | 240 | 24.0 | 2.8 | 6001 | 6.2 |
| 03.19.2019 | 1 | 20000 | 240 | 24.0 | 2.8 | 5001 | 6.35 |



**Table A2: Settings of the digital image correlation (DIC) analysis of the high-speed recordings. The DIC subsets were allowed to translate, rotate and deform with normal and shear. For the subset initialization, interpolation and optimization method we used the "field_values", "keys_fourth", and "gradient_based" setting of the DICengine software.**


| Date | # PST at day | Number of subsets | Subset stepsize (pixel) | Subset size (pixel) |
|---|---|---|---|---|
| 01.04.2019 | 1 | 17409 | 3 | 9 |
| 01.07.2019 | 3 | 31522 | 3 | 12 |
| 01.08.2019 | 1 | 27096 | 3 | 12 |
| 01.09.2019 | 1 | 29775 | 3 | 12 |
| 01.10.2019 | 1 | 28728 | 3 | 12 |
| 01.11.2019 | 1 | 27887 | 3 | 12 |
| 01.11.2019 | 2 | 35109 | 3 | 12 |
| 01.13.2019 | 1 | 46435 | 3 | 12 |
| 01.14.2019 | 1 | 46192 | 3 | 12 |
| 01.15.2019 | 1 | 38552 | 3 | 12 |
| 01.16.2019 | 1 | 38789 | 3 | 12 |
| 01.16.2019 | 2 | 26196 | 3 | 12 |
| 01.17.2019 | 1 | 23301 | 3 | 12 |
| 01.23.2019 | 2 | 20651 | 3 | 12 |
| 01.25.2019 | 1 | 19633 | 3 | 12 |
| 01.30.2019 | 1 | 46794 | 2 | 12 |
| 02.02.2019 | 1 | 49453 | 2 | 12 |
| 02.08.2019 | 1 | 49308 | 2 | 12 |
| 02.12.2019 | 1 | 53528 | 2 | 12 |
| 02.18.2019 | 1 | 51022 | 2 | 12 |
| 02.22.2019 | 1 | 20527 | 3 | 12 |
| 03.05.2019 | 1 | 18043 | 3 | 12 |
| 03.13.2019 | 1 | 20399 | 3 | 12 |
| 03.19.2019 | 1 | 20259 | 3 | 12 |





**Table A3: Camera and digital image correlation (DIC) settings of the Sony camera (RX100-V, 1920 x 1080 pixel$^2$ resolution). The DIC subsets were allowed to translate and rotate. For the subset initialization, interpolation and optimization method we used the "feature_matching", "keys_fourth", and "gradient_based" setting of the DICengine software.**

| Date | # PST at day | Frame rate (fps) | Number of subsets | Subset size (pixel) | Step size (pixel) | Threshold (-) | Frame rate (fps) | Pixel conversion (mm/pixel) |
|---|---|---|---|---|---|---|---|---|
| 01.04.2019 | 1 | 100 | 4077 | 11 | 4 | 60 | 100 | 1.49 |
| 01.07.2019 | 3 | 50 | 6611 | 21 | 6 | 60 | 50 | 1.85 |
| 01.08.2019 | 1 | 100 | 7125 | 25 | 4 | 30 | 100 | 2.41 |
| 01.09.2019 | 1 | 100 | 2156 | 21 | 6 | 60 | 100 | 2.53 |
| 01.10.2019 | 1 | 50 | 3852 | 23 | 6 | 60 | 50 | 2.26 |
| 01.11.2019 | 1 | 50 | 3316 | 21 | 6 | 60 | 50 | 2.35 |
| 01.11.2019 | 2 | 50 | 13228 | 21 | 5 | 30 | 50 | 1.72 |
| 01.13.2019 | 1 | - | - | - | - | - | - | - |
| 01.14.2019 | 1 | 50 | 7746 | 23 | 6 | 60 | 50 | 2.88 |
| 01.15.2019 | 1 | 50 | 5948 | 23 | 6 | 60 | 50 | 3.46 |
| 01.16.2019 | 1 | 50 | 6528 | 23 | 6 | 60 | 50 | 3.3 |
| 01.16.2019 | 2 | - | - | - | - | - | - | - |
| 01.17.2019 | 1 | 50 | 2313 | 23 | 6 | 30 | 50 | 5.38 |
| 01.23.2019 | 2 | 50 | 2661 | 21 | 5 | 60 | 50 | 5.47 |
| 01.25.2019 | 1 | 50 | 1841 | 19 | 5 | 60 | 50 | 5.41 |
| 01.30.2019 | 1 | 50 | 1969 | 21 | 6 | 60 | 50 | 5.49 |
| 02.02.2019 | 1 | 50 | 2695 | 21 | 6 | 60 | 50 | 5.27 |
| 02.08.2019 | 1 | 50 | 3506 | 23 | 5 | 60 | 50 | 5.28 |
| 02.12.2019 | 1 | 50 | 4243 | 21 | 5 | 60 | 50 | 5.35 |
| 02.18.2019 | 1 | 50 | 3958 | 23 | 5 | 40 | 50 | 5.2 |
| 02.22.2019 | 1 | 50 | 3103 | 21 | 5 | 30 | 50 | 5.23 |
| 03.05.2019 | 1 | 50 | 4495 | 21 | 4 | 30 | 50 | 5.09 |
| 03.13.2019 | 1 | 50 | 3701 | 21 | 5 | 30 | 50 | 4.75 |
| 03.19.2019 | 1 | 50 | 5043 | 21 | 4 | 30 | 50 | 4.88 |


**Appendix B: Stiffness properties of the slab derived with the layered slab model.**



**Figure B1: Stiffness quantities of the slab derived from the layered model with time. The stiffness quantities are (a) the extensional stiffness $A_{11}$, (b) the coupling stiffness $B_{11}$ which incorporates the bending-extension coupling of an asymmetrically layered slab, (c) the bending stiffness $D_{11}$ and (d) the shear stiffness $kA_{55}$. Error bars indicate the measurement uncertainty. Transparent dots indicate that the elastic modulus of the weak layer $E_{wl}$ did not converge within the boundaries given in the optimization routine, instead an upper boundary for $E_{wl}$ of 2 MPa was used.**

### Appendix C: Crack tip estimation.

Bobillier et al. (2021) reproduced the propagation saw test with a 3-D DEM simulation. They compared different techniques
to locate the crack tip during crack propagation in the weak layer. Their study showed that different crack tip definitions based
on displacement threshold (0.2 mm), on maximum stress or the position of breaking bonds are equivalent. From their
simulations we computed the downward acceleration of a beam section in the slab (Figure C1, top row) at different crack
propagation distances (Figure C1, columns) and compared it to the number of breaking bonds. The maximum of downward
acceleration occurred always at the time when the number of breaking bonds strongly increased. Therefore, the maximum of
acceleration can be seen as a further equivalent definition of the crack tip.

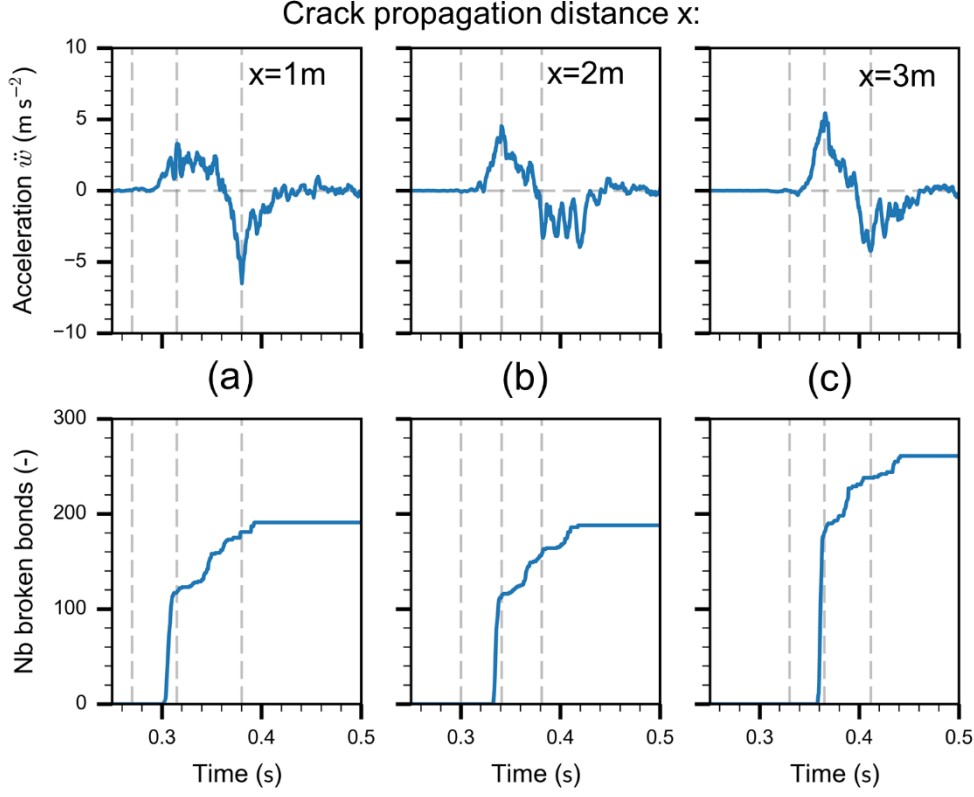

**Figure C1: (top row) Downward acceleration of the slab in a PST experiment with time. (bottom row) Cumulated number of broken
bonds between the discrete elements in the DEM simulation with time. The initial strong increase in broken bonds corresponds to
the crack tip. For all three crack propagation distances, the time of the steep increasing flank of bond breaking is well aligned with
the maximum of acceleration.**





**Appendix D: Snow cover modeling.**

The snow cover model SNOWPACK (version 3.60) was used to simulate the evolving snow stratigraphy at our field site (Bartelt and Lehning, 2002) (Figure D1a). The model was driven with data from an automatic weather station (AWS) which
was located 100 m next to the field site. From the weather station we used air temperature, relative humidity, snow surface temperature, wind speed, wind gust speed, wind direction and snow height (10 min averages). Short and long wave radiation data were obtained from another AWS, which is 1.7 km from the field site (IMIS station SLF 2). The influence of the surrounding terrain (e.g. shading and reflection) on the short wave radiation was accounted for using Alpine 3D (version 3.20) (Lehning et al., 2006). The snow cover mass balance was enforced with the increment of measured snow height from the
weather station at the field site. The SNOWPACK standard soil profile was modified to better fit our field site on the flat roof of a building (0.5 m concrete above 5 m air) and a constant heat flux of 0.005 W m$^{-2}$ was chosen. The simulation time step was one hour. The SNOWPACK initiation file will be made available on www.envidat.ch.

**Figure D1: (a) Temporal evolution of simulated snow stratigraphy. Colours are representing grain types as indicated in the colour bar on the right. (b) modelled crack speed (grey line) and modelled critical crack length, which were used to calculate SSP in section 4.4.**
