# Peer review of "Temporal evolution of crack propagation characteristics in a weak snowpack layer: conditions of crack arrest and sustained propagation"

_Natural Hazards and Earth System Sciences, 2022_

## Referee Comment (RC2)

**Temporal evolution of crack propagation characteristics in a weak snowpack layer: conditions of crack arrest and sustained propagation**

Bastian Bergfeld1, Alec van Herwijnen1, Grégoire Bobillier1, Philipp L. Rosendahl2, Philipp Weißgraeber3, Valentin Adam2,1, Jürg Dual4 and Jürg Schweizer1

120

[referee-annotated manuscript omitted]

---

## Author Comment (AC1)

Dear reviewer,

thank you very much for commenting and providing helpful suggestions on the manuscript. Below we have pasted your comments in blue, our point-by-point responses are given in black.

**Specific comments:**

First sentence of abstract: This sentence might seem a bit confusing and complicated for readers not familiar with slab avalanche release. I suggest making it simple: "For a slab avalanche to release, we need crack propagation in a weak snow layer beneath a cohesive snow slab." Then continue describing crack propagation, etc.

Thanks for the suggestions. We will change this sentence so that it becomes clearer.

Abstract again: a PST is not common knowledge, I would think. Maybe write "we performed crack propagation experiments… ". You can then describe the PST in more detail in the intro and/or methods section.

We will reword the sentence and not introduce already the PST in the Abstract.

Figure 2: What about the crack speed curves for the other experiments? It would be interesting to see the others as well. Are they very similar or completely different? If there are differences, how can those differences be explained?

The crack speed curves of experiments resulting in full propagation had similar shapes (see figure below).

[Figure]

We think the data shown in Figure 2 is representative. Experiments resulting in crack arrest showed greater differences and larger uncertainty. We will make these data available with publication at our institutional repository [www.envidat.ch](www.envidat.ch).

Line 225: How valid is the assumption that the slab and substratum are in the same stress state before and after crack propagation? Will there be no plastic deformation within the slab due to the collapse?

Admittedly, we cannot easily verify this assumption. However, within the limits of our measurement accuracy we did not detect residual strain in the slab and substratum (with the exception of the explicitly mentioned experiment in line 275 – see next comment). In the revised manuscript we will state this issue more clearly.

Line 275: Here you say that the slab was shallow and soft and it broke while cutting the weak layer. How does this fit with the assumption in line 225?

We only observed residual strain in the slab in the experiment of 4 January 2019. Hence, the assumption made in line 225 does not hold for this particular experiment. However, this experiment did not result in crack propagation, and we did not use this experiment to derive crack propagation characteristics such as fracture energies.

Discussion: In general, the first sentence of a paragraph should summarize the paragraph and tell the reader what the paragraph is about. I have the impression that this "first sentence-summary" concept was not used in the discussion, which makes it a bit tedious to read. I suggest adding "first-sentence-summaries" at the beginning of the paragraphs.

Thanks, for the suggestion. We will follow your advice in the revised manuscript where we deem it necessary.

**Technical corrections**

Line 52 and at many further places throughout the document: Variable identifiers (in our case the f and the dyn) are not italicized.

We will change this in the revised manuscript.

Line 78: omit the "initial"

We will change as suggested.

Lines 84-85: firstly and secondly (adverbs referring to the verb "is")

First and second can be used as adjectives as well as adverbs.

Figure 1b: using a second y-axis (on the right) for the temperature would be more elegant.

We will change as suggested.

---

## Author Response (AR1)

Dear reviewer,

thank you very much for commenting and providing helpful suggestions on the manuscript. Below we have pasted your comments in blue, our point-by-point responses are given in black.Line numbers refer to the revised manuscript with "track changes".

**Specific comments:**
First sentence of abstract: This sentence might seem a bit confusing and complicated for readers not familiar with slab avalanche release. I suggest making it simple: "For a slab avalanche to release, we need crack propagation in a weak snow layer beneath a cohesive snow slab." Then continue describing crack propagation, etc.

Thanks for the suggestions. We changed this sentence so that it becomes clearer (line 12).

Abstract again: a PST is not common knowledge, I would think. Maybe write "we performed crack propagation experiments… ". You can then describe the PST in more detail in the intro and/or methods section.

We reworded the sentence and now do not introduce the PST already in the Abstract (lines 14 and 18)

Figure 2: What about the crack speed curves for the other experiments? It would be interesting to see the others as well. Are they very similar or completely different? If there are differences, how can those differences be explained?

The crack speed curves of experiments resulting in full propagation had similar shapes (see figure below).

[Figure]

We think the data shown in Figure 2 is representative. Experiments resulting in crack arrest showed greater differences and larger uncertainty. We make these data available with publication at our institutional repository [www.envidat.ch](www.envidat.ch).

Line 225: How valid is the assumption that the slab and substratum are in the same stress state before and after crack propagation? Will there be no plastic deformation within the slab due to the collapse?

Admittedly, we cannot easily verify this assumption. However, within the limits of our measurement accuracy we did not detect residual strain in the slab and substratum (with the exception of the explicitly mentioned experiment in line 280 – see next comment). In the revised manuscript we now state this issue more clearly in line 231.

Line 275: Here you say that the slab was shallow and soft and it broke while cutting the weak layer. How does this fit with the assumption in line 225?

We only observed residual strain in the slab in the experiment of 4 January 2019. Hence, the assumption made in line 230 does not hold for this particular experiment. However, this experiment did not result in crack propagation, and we did not use this experiment to derive crack propagation characteristics such as fracture energies. In this context, we made a correction in line 339, which now excludes the experiment of 4 January 2019.

Discussion: In general, the first sentence of a paragraph should summarize the paragraph and tell the reader what the paragraph is about. I have the impression that this "first sentence-summary" concept was not used in the discussion, which makes it a bit tedious to read. I suggest adding "first-sentence-summaries" at the beginning of the paragraphs.

Thanks, for the suggestion. We followed your advice in the revised manuscript where we deemed it necessary.

**Technical corrections**

Line 52 and at many further places throughout the document: Variable identifiers (in our case the f and the dyn) are not italicized.

We changed this in the revised manuscript.

Line 78: omit the "initial"

We changed as suggested.

Lines 84-85: firstly and secondly (adverbs referring to the verb "is")

First and second can be used as adjectives as well as adverbs; hence, we prefer to keep first and second.

Figure 1b: using a second y-axis (on the right) for the temperature would be more elegant.

We changed as suggested.

Dear Dr. Bair,

thank you for the positive feedback and useful comments which will improve the manuscript. Below, your comments are in blue, our point-by-point responses are given in black. Line numbers refer to the revised manuscript with track changes.

1. The authors note that in some cases the "snowpack tended to arrest cracks without slab fractures" but no information is provided about how the absence of slab fracture was measured.

Thanks for pointing this out, in the revised mansucript we now mention that we did not observe strain concentrations in the slab that are typically associated with slab fractures (line 231).

2. The choice of column length in all the experiments (Table 1) is not justified. I presume other tests were done before the authors decided to excavate a 9 m long PST?

The choice of column length was mainly restricted by logistical constraints. The width of the roof (our field site) was 10 meters, while the length was 30 meters. To film such long PST experiments, the distance between the camera and the side wall of the PST has to be at least 12 meters. As initially we did not want to remove a large part of the snow, we used shorter columns for the first few experiments. Once we had removed enough snow to allow full width experiments, we always performed 9 m long PST's.

The abstract says that tests up to 10 m long were performed, but the longest column length in Table 1 is 9 m. Please explain.

This is an error. The width of the roof was 10 m but the tests were somewhat shorter. We corrected this in the revised manuscript (line 14).

3. Edge effects from the near and far end of the PSTs are discussed, but the edge effect of the width of the PST is only briefly mentioned (as an experimental error on Feb 22 2019). I assume the width of most of the PSTs was 30 cm? This needs to be stated.

We now state the PST column width of 30 cm in the revised manuscript (line 96)

4. There are at least 2 studies, e.g., Bobillier (2022) and Trottet et al. (2022), that are not publicly available, as they are in review and in press, respectively. By Copernicus (standards https://publications.copernicus.org/for_authors/manuscript_preparation.html), these articles can only be cited if they are available to reviewers, and must be publicly available at the time of final submission.

The study by Trottet et al. (2022) is now published and available under the following DOI: https://doi.org/10.1038/s41567-022-01662-4

The study from Bobillier et al. (2022) is still under review. I sent you the current version by e-mail on 5 Nov 2022.

**Below we comment on the annotated PDF:**

LINE 37: Of 10 crowns carefully examined in situ or photographically, almost all were thinner at the flanks, suggesting that arrest has a strong dependence on slab thickness. See Bair et al (2010).
Bair, E. H., K. W. Birkeland, and J. Dozier (2010), In situ and photographic measurements of avalanche crown transects, Cold Regions Science and Technology, 64(3), 174-181, doi: 10.1016/j.coldregions.2010.08.004.

Thanks for pointing out this study. We agree that slab thickness can influence crack propagation. Here we wanted to point out that crack arrest can occur if snowpack changes are gradual and even if no clear snowpack changes are observed.

LINE 98: Any other important characteristics about this site that made it an ideal outdoor laboratory? Does it get direct sunlight that time of year? Any wind? Is the building heated? Is the roof concrete?

We now describe the site in more detail in the revised manuscript (lines 97 – 100). The site is surrounded by trees which protect it from wind. The nearby creek, together with the cold concrete roof (snow-concrete interface usually colder than -5 °C) promote the growth of surface hoar, and direct sunlight does not reach the field site until the end of February. These factors make the site an ideal outdoor laboratory for large crack propagation experiments under controlled snowpack conditions.

LINE 100: Is that the layer thickness or the crystal size, or both?

It is both. We clarified this in line 101 of the revised manuscript.

FIGURE 1: That's a thick red area. Did the surface hoar layer develop over multiple days?

Indeed, the surface hoar developed during the period indicated by the red area. It was a period of cold, clear weather without precipitation and wind. We now mention this in the revised manuscript (line 102).

LINE 131: Can you expand on this? If tracking a saw with a black dot against a white background automatically is unreliable, then how can the DIC tracking be reliable for the snow?

Tracking the black dot mounted on the saw was unreliable as the algorithm sometimes did not find the dot in the image. This was caused by light reflections, as the dot was printed on paper and laminated in plastic, strongly reflecting light. For the manual snow saw tracking, this was not an issue.

LINE 141: exp is experimental?

Yes, we clarified this in the revised manuscript in line 143.

LINE 142: subscripts needed

Thank you for catching this error. We corrected this in the revised manuscript.

LINE 183: How was the area free from edge effects defined? It appears as the acceleration/deceleration is below some unstated threshold.

That is correct, for mean speed computation, we choose the propagation range where the crack speed showed less acceleration/deceleration. We did not use a fixed threshold value, but selected this range subjectively by hand.

LINE 269: Maybe put a reference to this section earlier, when the manual section is introduced without any further justification. Also there are still edge effects from the limited width of the beam that should be acknowledged.

This is related to question 3 above. As long as the column width does not change (in horizontal and vertical direction) we are not aware of effects affecting our measurements.

TABLE 1: How was the column length selected? Were other tests done first?

Please refer to our reply on your question 2 above.

LINE 297: That statement could be generalized to most of the results. PSTs on flat terrain are dominated by vertical displacement.

True, as we stated in line 302, flat field PSTs are always dominated by vertical displacements. At this point, we deem it worth to stay very specific to discuss the implications (e.g. measuring a flexural elastic modulus rather than a Young's modulus)

LINE 417: what ? authors? studies?

We clarified this in the revised manuscript (Line 424).

LINE 486: These 2 studies are in review and in press respectively, and neither are available publicly. Please fix and make sure they are publicly available for final submission

Please see our reply to your question 4 above.

TABLE A2: This table is unnecessary and could be summarized in a sentence or two.

To reproduce the results, this table is necessary. We do not see how all values can be mentioned in two sentences. We agree that most readers will not need this information, which is the very reason we have placed it in the Appendix.